# Immunoinformatics Analysis of SARS-CoV-2 ORF1ab Polyproteins to Identify Promiscuous and Highly Conserved T-Cell Epitopes to Formulate Vaccine for Indonesia and the World Population

**DOI:** 10.3390/vaccines9121459

**Published:** 2021-12-09

**Authors:** Marsia Gustiananda, Bobby Prabowo Sulistyo, David Agustriawan, Sita Andarini

**Affiliations:** 1Department of Biomedicine, School of Life Sciences, Indonesia International Institute for Life Sciences, Jl. Pulomas Barat Kav 88, Jakarta 13210, Indonesia; bobby.sulistyo@i3l.ac.id; 2Department of Bioinformatics, School of Life Sciences, Indonesia International Institute for Life Sciences, Jl. Pulomas Barat Kav 88, Jakarta 13210, Indonesia; david.agustriawan@i3l.ac.id; 3Department of Pulmonology and Respiratory Medicine, Faculty of Medicine University of Indonesia, Persahabatan Hospital, Jl Persahabatan Raya 1, Jakarta 13230, Indonesia; sita.laksmi@ui.ac.id

**Keywords:** SARS-CoV-2, immunoinformatics, T-cell epitopes, multi-epitope peptide-based vaccine, cytotoxic T-cells, helper T-cells, human leukocyte antigen, HLA-A*24:07

## Abstract

SARS-CoV-2 and its variants caused the COVID-19 pandemic. Vaccines that target conserved regions of SARS-CoV-2 and stimulate protective T-cell responses are important for reducing symptoms and limiting the infection. Seven cytotoxic (CTL) and five helper T-cells (HTL) epitopes from ORF1ab were identified using NetCTLpan and NetMHCIIpan algorithms, respectively. These epitopes were generated from ORF1ab regions that are evolutionary stable as reflected by zero Shannon’s entropy and are presented by 56 human leukocyte antigen (HLA) Class I and 22 HLA Class II, ensuring good coverage for the Indonesian and world population. Having fulfilled other criteria such as immunogenicity, IFNγ inducing ability, and non-homology to human and microbiome peptides, the epitopes were assembled into a vaccine construct (VC) together with β-defensin as adjuvant and appropriate linkers. The VC was shown to have good physicochemical characteristics and capability of inducing CTL as well as HTL responses, which stem from the engagement of the vaccine with toll-like receptor 4 (TLR4) as revealed by docking simulations. The most promiscuous peptide ^899^WSMATYYLF^907^ was shown via docking simulation to interact well with HLA-A*24:07, the most predominant allele in Indonesia. The data presented here will contribute to the in vitro study of T-cell epitope mapping and vaccine design in Indonesia.

## 1. Introduction

The COVID-19 disease caused by SARS-CoV-2 (severe acute respiratory syndrome coronavirus 2) has become a pandemic with dramatic socioeconomic consequences [1,2]. As of 21 September 2021, around 228 million people have been infected and approximately 4.6 million deaths have been reported worldwide (https://covid19.who.int/ accessed on 22 September 2021) [3]. In Indonesia alone, as of 26 September 2021, there have been around 4.2 million confirmed cases with 141,381 deaths (https://covid19.go.id/peta-sebaran accessed on 27 September 2021) [4]. The virus was first identified in Wuhan, China, and based on the sequence similarity, was thought to have originated from BatCov RaTG13 [5]. Like any other viral disease, individuals with COVID-19 might have varied symptoms, such as fever or chills, cough, fatigue, muscle aches, headache, or diarrhea. The severity of the symptoms is quite broad, and based on severity; the NIH has classified COVID-19 into five distinct types, namely, asymptomatic, mild, moderate, severe, and critical illness [6]. Patients with severe respiratory illness or acute respiratory distress syndrome might require intensive care and intubation, and this frequently may lead to death. Age and the presence of underlying comorbidities seem to determine the course and outcome of the diseases [7].

SARS-CoV-2 is a positive single-stranded RNA virus with a genome size of 30 kilobases (kb) containing 10 open reading frames (ORFs) that encode for structural, non-structural, and accessory proteins. The largest open reading frame of SARS-CoV-2 is called ORF1ab, occupying the second third of the genome, and encodes a replicase polyprotein 1ab (7096 amino acids). Polyprotein 1ab must undergo processing by the virally encoded proteases known as the chymotrypsin-like protease (3CLpro, NSP5) and the papain-like protease (PLpro, NSP3). Both proteases are initially part of the Polyprotein 1ab before they were autocatalytically cleaved from the strand and released. The proteases then cleave the other proteins from the Polyprotein 1ab into a total of 16 non-structural proteins, which are involved in replication and transcription of viral genome [8,9]. The rest of the genome (10 kb) encodes for five structural proteins, namely, nucleocapsid (N), membrane (M), surface (S), and envelope (E) proteins and accessory proteins, namely, ORF3, ORF6, ORF7, ORF8, and ORF10 [10].

Vaccines are one of the most important countermeasures against the dire consequences of SARS-CoV-2 spread in the human population. Current vaccines are quite effective in controlling mortality, morbidity, and hospitalization related to COVID-19. Many of the vaccines are aiming to induce the response of antibodies against the spike glycoproteins, which will lead to blockage of viral entry into the cells. However, the emergence of the new variants raises concerns about the long-term effectiveness of the vaccines and escape from antibody detection [11]. SARS-CoV-2 variants emerged due to the high mutation rates of the RNA viruses pertaining to the low fidelity of the RNA-dependent RNA polymerase (RdRp). In SARS-CoV-2, high mutation frequency was observed for the S protein. Some of the variants have caused significantly higher fatality rates in some countries [12]. These variants have a D614G mutation in the spike proteins which was hypothesized to increase the infectivity of the SARS-CoV-2 virus [13]. As of 23 September 2021, the Centre for Disease Control and Prevention (CDC) has designated these virus variants into four categories, namely, Variant Being Monitored (VBM), which includes Alpha (B.1.1.7, Q.1-Q.8), Beta (B.1.351, B.1.351.2, B.1.351.3), Gamma (P.1, P.1.1, P.1.2), Epsilon (B.1.427 and B.1.429), Eta (B.1.525), Iota (B.1.526), Kappa (B.1.617.1, B.1.617.3), Mu (B.1.621, B.1.621.1), and Zeta (P.2); Variant of Concern (VOC) which include Delta (B.1.617.2 and AY.1 sub-lineages); Variant of Interest (VOI); and Variant of High Consequence (VOHC). At present, no SARS-CoV-2 variants are categorized as VOI and VOHC (https://www.cdc.gov/coronavirus/2019-ncov/variants/variant-info.html accessed on 24 September 2021) [14].

In a situation where antibodies failed to block viral entry into the cells, the other arm of adaptive immunity, namely cell-mediated immunity mediated by CTL, is needed to curb the infection and control the diseases to subclinical level. Inside the infected cells, viral proteins will undergo HLA Class I pathway, where they will be firstly tagged by ubiquitin and then digested into short peptides by the proteasome. These short peptides (8–10 amino acids long) are then translocated with the help of transporter associated with antigen processing (TAP) into the endoplasmic reticulum (ER) where HLA proteins are translated. Peptide binds to the HLA molecule and complex is presented on the surface of the infected cell to be scrutinized by T-cells. Upon recognizing the complex, T-cells kill the infected cells. The peptides, termed T-cell epitopes, are 8–10 amino acids long and can originate from any viral proteins, not only the spike glycoprotein. Because T-cell recognizes an antigen as a complex with HLA molecules, T-cell immunity, therefore, depends on the HLA molecules that an individual has. In humans, HLA proteins are encoded by the HLA gene, which is considered the most variable gene within the human genome. In light of the situation where we expect that the new SARS-CoV-2 variants might occur, identification of T-cell epitopes that originate from conserved regions of the virus, and utilizing them for vaccines, should be a priority.

A peptide-based vaccine that induces cell-mediated immunity is a likely choice for emerging virus vaccines, including vaccines for SARS-CoV-2. Viral spike protein is more likely to mutate, and vaccines against it might not be able to prevent the infection. In that situation, having a good T-cell memory that can identify and eliminate infected cells will be beneficial and could potentially save lives. There have been several peptide-based vaccines for COVID-19 in the pipeline undergoing pre-clinical and clinical trials (https://www.who.int/publications/m/item/draft-landscape-of-COVID-19-candidate-vaccines accessed on 25 September 2021) [15]. Several peptides-based vaccines are composed of epitope peptide pool from the spike receptor binding domain (NCT04545749, NCT04683224), spike, and nucleoprotein (NCT04780035), and all proteins of SARS-CoV-2 (NCT04954469, NCT04885361).

ORF1ab, being the largest ORF in the SARS-CoV-2 genome could become the main source of T-cell epitopes. ORF1ab is the first protein to be translated by the infected cell making ORF1ab a good source of early T-cell responses. ORF1ab is also quite stable genetically, with only a small number of mutations in the protein sequences detected when compared to the ancestral sequence of Wuhan Hu-1 [16]. Therefore, in this study, we focused on the identification of conserved and promiscuous CTL and HTL epitopes from ORF1ab to design a multi-epitope peptide-based vaccine that will cover a large portion of human population and be effective against any SARS-CoV-2 variants.

## 2. Materials and Methods

The overall methodology and protocol for the identification of CTL and HTL epitopes from SARS-CoV-2 ORF1ab is illustrated in Figure 1.

### 2.1. SARS-CoV-2 ORF1ab Sequence Retrieval

ORF1ab protein sequence of SARS-CoV-2 Wuhan Hu-1 was retrieved from the reference sequence in NCBI (YP_009724389.1). The reference sequence was used in the prediction of T-cell epitopes. The other SARS-CoV-2 ORF1ab sequences were retrieved from NCBI Virus SARS-CoV-2 Data Hub (https://www.ncbi.nlm.nih.gov/labs/virus/vssi/#/SARS-CoV-2) accessed on 22 September 2021. All sequences were retrieved in FASTA format and used for the epitope conservancy analysis. ORF1ab sequences were selected using predefined filters: sequence length 7093–7096 amino acids, the maximum number of ambiguous characters set to 0, human for the host, Pango lineages chosen were Alpha (B.1.1.7), Beta (B.1.351), Delta (B.1.617.2), Eta (B.1.525), Gamma (P.1), Iota (B.1.526), Kappa (B.1.617.1), Lambda (C.7), and Mu (B.1.621), and the completeness option for sequences was set to complete. Due to the large number of delta sequences available in the database, only the isolates from 31 July to 22 September 2021 were included.

### 2.2. Entropy Analysis of 9-Mer Peptide Sequences

To assess the degree of conservation and variability of 9-mer sequences representing T-cell epitopes within ORF1ab, we calculated Shannon’s entropy according to the methods described in Khan et al. (2017) [17]. Following retrieval of SARS-CoV-2 ORF1ab protein sequences, duplicates were removed using AliView [18] to avoid bias in the calculation of entropy. We then conducted multiple sequence alignment of the remaining sequences using MAFFT v.7.144 (10.1093/molbev/mst010) available on CIPRES portal (https://www.phylo.org/ accessed on 28 October 2021) [19], with default alignment setting. The finished alignment was used as input for entropy analysis using AVANA [20], with sample size set to 9 for 9-mers, entropy values extrapolated to infinite sets with 100 random subalignment sampling to correct for size bias, and highly gapped positions were defined as positions where gaps are 50% of the symbols. The statistics for variability were then imported and processed in Microsoft excel, where the entropy values were plotted against the 9-mer center positions.

### 2.3. Retrieval of HLA Alleles Type in INDONESIAN Population as the Bases for Prediction

The HLA allele types of Indonesian population were retrieved from The Allele Frequency Net Database (AFND) (http://www.allelefrequencies.net accessed on 21 September 2021) [21]. Most of the HLA alleles data for the Indonesian population listed in AFND came from one study conducted in the Javanese and Sundanese Javanese populations [22]. The data can be considered representative of the Indonesian population, as among 300 distinct ethnic and linguistic groups that exist in Indonesia, Javanese, and Sundanese Javanese are the largest ethnic population, accounting for 40% and 15.5%, respectively (world population review, accessed on 1 October 2021) (https://worldpopulationreview.com/countries/indonesia-population, accessed on 21 September 2021) [23].

### 2.4. Retrieval of the Number of Experimentally Validated ORF1ab Epitopes Associated with Predominant Indonesian HLA Alleles

Cell-mediated immunity to SARS-CoV-2 has been a topic of interest for many researchers in the last 18 months since the pandemic arose, and hence several identified T-cell epitopes are already available in the Immune Epitope Database (IEDB) [24]. For each HLA allele (with allele frequency > 5%) that the Indonesian population has, we retrieved the information about how many positive T-cell epitopes associated with the alleles are already reported in IEDB (IEDB accessed on 18 September 2021) either by T-cell assay or HLA assay. We then further noted whether the HLA allele is specific to Indonesia or shared with Germany, as representative of the Caucasian population and Thailand as representative of other Southeast Asia population. The HLA alleles for Germany were retrieved from the SARS-CoV-2 T-cell epitopes identification study [25] and HLA alleles of the Thai population were retrieved from a large-scale HLA typing study [26,27,28] focusing only the HLA alleles with a minimum 5% frequency in the Thai population.

### 2.5. Prediction of CTL Epitopes from ORF1ab

The 9-mer peptides from ORF1ab were analyzed for their capacity to enter the HLA Class I presentation pathway and become T-cell epitopes that will be recognized by CTL. Immunoinformatics server NetCTLpan 1.1 was used in the analysis (https://services.healthtech.dtu.dk/service.php?NetCTLpan-1.1, accessed on 21 September 2021) [29]. The server calculated all steps involved in the HLA Class I antigen processing and presentation pathways, such as the efficiency of proteasomal cleavage, efficiency of the transporter associated with antigen processing (TAP) to translocate peptides from the cytosol into the endoplasmic reticulum (ER) lumen, and finally predict the binding affinity of peptide to HLA molecules. The parameters for the weight placed on C-terminal cleavage and the antigen transport efficiency were set to the default values of 0.225 and 0.025, respectively. Stringent selection criteria were applied, where only the top 1% rank peptides were considered as CD8^+^ T-cell epitopes and subjected to further analysis. The percentile rank is proportionally related to the binding affinity of the peptide to the HLA molecule. Therefore, the low percentile rank applied here will ensure that only peptides with a higher likelihood to become the real CTL epitopes will be selected. NetCTLpan is the sole prediction server used, as it was developed based on the NetMHCpan method, which was shown to be the best predictor for peptide–HLA binding [30], and it can predict peptide binding to HLA molecules even though the experimental data are not available.

### 2.6. Prediction of HTL Epitopes from ORF1ab

Immunoinformatics server NetMHCIIpan 4.0 (https://services.healthtech.dtu.dk/service.php?NetMHCIIpan-4.0, accessed on 21 September 2021) [31] was used to predict the binding affinity of the 15-mer peptides to HLA Class II alleles. HLA class II molecules, due to the open conformation of the peptide-binding groove, can accommodate longer peptides, with additional 3 residues on each flank. The binding core of the interaction, however, consists of the 9-mer peptide. Both 15-mer and 9-mer binding core were curated. In this analysis, strong and weak binding is defined by the percentile rank thresholds of 1% and 5%, respectively. The peptides that belong to the strong binders were selected and subjected to further analysis.

### 2.7. Immunogenicity Analysis of Predicted CTL Epitopes

Binding affinity of peptide to HLA molecule does not determine immunogenicity. Immunogenicity of peptides is determined by the presence of T-cells having T-cell receptor recognizing the peptide-HLA complex [32]. CD8^+^ T-cells recognize peptide antigens as a complex with HLA Class I molecules. Some residues of the peptides (positions 1, 2, and 9) bind to the peptide-binding groove of HLA molecules, and some other residues (positions 3–8) bind to the T-cell receptor. Immunogenicity analysis was conducted using the IEDB immunogenicity tool (http://tools.iedb.org/immunogenicity/, accessed on 21 September 2021) [33]. The higher the score generated by the tool, the more immunogenic the peptide is. In general, the presence of large and aromatic residues is associated with immunogenicity, and residues number 4–6 of the presented peptides was shown to have a large effect on immunogenicity [33]. In this study, all CTL epitopes from ORF1ab peptides chosen in the preceding step were subjected to immunogenicity analysis applying the parameters where residue no. 1, 2, and C-terminal were masked. The peptides with positive immunogenicity scores were selected for further analysis.

### 2.8. Interferon-Gamma (IFNγ)-Inducing Ability of Predicted HTL Epitopes

The HTL is important for the generation of the cytokines that drive appropriate immune responses. For intracellular pathogens such as viruses, IFNγ is a very important cytokine for CD8^+^ T-cell differentiation into a full effector CTL and memory CTL. HTL come in different subsets, and the subsets that will produce IFNγ were named Th1. Therefore, the ability of HTL epitopes to induce the production of IFNγ was analyzed using the IFNepitope server (http://crdd.osdd.net/raghava/ifnepitope/scan.php, accessed on 21 September 2021) [34]. The parameter for prediction was set as default using the supportive vector machine approach, which will calculate the score for each peptide likelihood to induce IFNγ. The peptides having positive IFNγ scores were selected to be included in the next analysis step.

### 2.9. Conservancy Analysis of the Predicted Epitopes against SARS-CoV-2 Variants

Both selected CTL and HTL epitopes were subjected to conservancy analysis using the IEDB epitope conservancy analysis tool (http://tools.iedb.org/conservancy/, accessed on 21 September 2021) [35] against SARS-CoV-2 variants. The duplicated sequences were removed before the analysis to avoid bias. Epitopes having conservancy level 100% across SARS-CoV-2 variants were short-listed to be included in the next step analysis.

### 2.10. Validation of Predicted Epitopes in IEDB Epitopes List

IEDB contains the experimentally known epitopes from the ORF1ab SARS-CoV-2 as well as the entire proteome as depicted in Table 1. The list of experimentally known epitopes can serve as a means to partially validate the T-cell epitope prediction. The ORF1ab peptides were curated using epitope search tools in the Immune Epitope Database (www.IEDB.org, accessed on 21 September 2021) [24], specifying the name of the pathogen (SARS-CoV-2), antigen (ORF1ab), and the host (human). The predicted peptides that matched with the experimentally validated peptides were prioritized to be included in the VC.

### 2.11. Cross-Reactivity of Predicted Epitopes with Human Peptides

The viral peptides having sequences that match with the sequence of the human self-peptides might induce either autoreactive T-cells or tolerogenic T-cells. The first one will cause an autoimmune response while the latter will reduce the immunogenicity of the vaccine. BlastP analysis (http://blast.ncbi.nlm.nih.gov/Blast.cgi, accessed on 21 September 2021) was conducted to find 9-mer peptide sequences within the human (taxid: 9606) proteome that match with the 9-mer sequences derived from SARS-CoV2, since both HLA Class I and Class II have peptide-binding core regions that can accommodate peptides with a 9 amino acid length. The BlastP algorithm parameter was set as follows: expect threshold 30,000, word size 2, matrix PAM30, gap cost was set to existence = 9 and extension = 1, the compositional parameter was set to no adjustment, and the low complexity filter was disabled and automatically adjusted for short input sequences. The results from BlastP analysis were transferred into Microsoft Excel and were screened for the peptides that shared at least contiguously 7 identical amino acid residues with the human peptides with no gap and no mismatches residue.

### 2.12. Epitope Selection and Vaccine Construction

Selection of the epitopes to be included in the VC followed several criteria: (1) Epitopes should be promiscuous so that they can be presented by many HLA alleles and hence generate a high-population coverage of the vaccine. (2) Epitopes presented by HLA Class I should be immunogenic so that there will be T-cells within the repertoire that will be able to respond to the peptides. (3) Epitope presented by HLA Class II should be able to induce IFNγ responses so that vaccine will be able to activate the Th1 responses that are needed for antiviral immune responses. (4) Epitopes should not have homology with the human peptides so that autoimmune responses triggered by the vaccine can be avoided while ensuring the immunogenicity of the VC.

The vaccine was designed by joining individual epitopes into a polypeptide. β-defensin was used as an adjuvant and will serve as a ligand for the TLR4 that is needed for dendritic cell maturation and successful T-cell activation in the lymph node. β-defensin, HTL, and CTL epitopes were connected using linkers EAAAK, GPGPG, and AAG, respectively [36].

### 2.13. Evaluation of VC Properties: Antigenicity, Allergenicity, Toxicity, and Physicochemical Characteristics

Antigenicity prediction was conducted using Vaxijen (http://www.ddg-pharmfac.net/vaxijen/VaxiJen/VaxiJen.html, accessed on 21 September 2021) [37] with the chosen parameter as follows: virus as target organism and score 0.4 as the antigenicity threshold.

Allergenicity analysis of the VC was conducted using two servers, namely Allertop 1.0 (http://www.pharmfac.net/allertop/, accessed on 21 September 2021) [38] and AllergenFP v.1.0 (https://ddg-pharmfac.net/AllergenFP/index.html, accessed on 21 September 2021) [39]. Allertop 1.0 method was trained using a set of equal number of known allergens (2210) and non-allergens (2210) from the match species. AllergenFP v.1.0 was trained using a set of 2427 allergens and 2427 non-allergens which are assembled into a matrix for prediction.

Physicochemical characteristics of the VC such as amino acid composition, molecular weight, pI, half-life, stability, and grand average of hydropathicity (GRAVY) were evaluated using Protparam tools (https://web.expasy.org/cgi-bin/protparam/protparam, accessed on 21 September 2021). The tools deduced these properties from a protein sequence [40]. These properties need to be considered for successful manufacturing process of the VC.

The possibility that the VC will generate toxic peptides was evaluated using Toxinpred (https://webs.iiitd.edu.in/raghava/toxinpred/index.html, accessed on 21 September 2021) [41]. The module Protein Scanning was used to generate all possible overlapping 10-mer peptides and predict the toxicity of the peptides using the SVM (Swiss-Prot)-based method with the threshold set to 0.0.

### 2.14. Re-Analyze the VC for Epitopes Generation and Homology with Human Proteins and Human Microbiome

The VC was analyzed to recheck that the CTL and HTL epitopes, which were put into the construct, will be generated, and that the other 9-mer peptides that might be generated do not have homology with human peptides and human microbiomes. CTL and HTL epitopes prediction were conducted using NetCTLpan1.1 and NetMHCIIpan4.0, respectively. BlastP was used to check for the peptide homology with human peptides.

The similarity of T-cell epitopes with the human microbiome might either dampen or increase immunogenicity [42]; therefore, it is important to validate that the vaccine will not disrupt immune homeostasis in the gut. A possibility that vaccine construct might generate epitopes that are homologous to the epitopes from the human microbiome was checked using Pipeline Builder for Identification of drug Targets (PBIT) server (http://www.pbit.bicnirrh.res.in/index.php, accessed on 21 September 2021) [43]. Predicted peptides were submitted as FASTA files to the PBIT server to check for sequence identity with the peptide from human microbiome. The peptides would be considered as non-homologous if the sequence identity was <50% and e-value was >0.005.

### 2.15. Immune Simulation of the VC

Immunological responses generated by the VC were assessed in silico using C-ImmSim online server (https://kraken.iac.rm.cnr.it/C-IMMSIM/, accessed on 21 September 2021) [44]. The simulation was conducted for the highest frequency of HLA haplotype found in the Indonesian population namely haplotype HLA-A*3401, HLA-B*1521, HLA-DRB1*1502 (4.6%), and HLA-A*2407, HLA-B*3505, HLA-DRB1*1202 (4.3%) [22]. In order to simulate prime-boost-boost vaccination, the simulation was run for a total of 1000 phases with three injections of 1000 units of vaccine that were given at an interval of four weeks apart (day 0, 28, and 56) that correlated with 1, 84, and 168 time-steps parameters in the simulation server. Note that 1 time-step is equal to 8 h and the injections were administered four weeks apart [45].

### 2.16. Population Coverage of the VC

The population coverage of the VC was evaluated using the population coverage analysis tool housed in IEDB (http://tools.iedb.org/population/, accessed on 21 September 2021) [35]. The population coverage analysis was conducted to ensure that the T-cell epitope-based vaccine will cover a large population. T-cell recognizes peptides presented by HLA molecules, and different ethnicities will express HLA types at different frequencies. The predicted population coverage represents the percentage of individuals that will respond to the VC and generate an immune response.

### 2.17. Secondary Structure and Tertiary Structure Prediction of the VC

The secondary structure of the VC was predicted using SOPMA (https://npsa-prabi.ibcp.fr/cgi-bin/npsa_automat.pl?page=/NPSA/npsa_sopma.html, accessed on 21 September 2021) [46], which is one of the automated methods of protein secondary structure prediction from multiply aligned protein sequences. RAPTOR X was used to validate the secondary structure and predict the tertiary structure of the VC (http://raptorx.uchicago.edu/ContactMap/, accessed on 21 September 2021) [47,48]. RAPTOR X predicts the tertiary structure based on the inter-residue distance distribution of a protein by the deep learning method. The server works best for predicting protein structures that do not have many sequences homology, and it has proven to be the best server for contact prediction that can be run using a personal computer [49]. Tertiary structure generated by RAPTOR X was then validated by using ProSA-web (https://prosa.services.came.sbg.ac.at/prosa.php, accessed on 21 September 2021) [50]. The PDB structure generated by RAPTOR X was used as an input file for ProSA-web, which will compare the predicted 3D model of the VC with the existing proteins structure in PDB database that were generated experimentally either by X-ray crystallography or NMR. ProSA-web then calculated the z-score, which represents the quality index of the model. Graphically, the z-score value is displayed in a plot that contains the z-scores of all experimentally determined protein chains in the PDB database, where dark blue and light blue area represents NMR and x-ray structures, respectively. The z-score value of the predicted model is displayed as a black dot in the graph, and the model quality is acceptable if it falls within the range of scores typically found for native proteins of similar size.

### 2.18. Molecular Docking of the VC with TLR4

The interaction between the vaccine and TLR4 (PDB ID: 3FXI) was modeled using HDOCK (http://hdock.phys.hust.edu.cn/, accessed on 21 September 2021) [51,52]. HDOCK generates information about the interacting residues between TLR4 and the vaccine. The PDB file of TLR4 (3FXI) and PDB file of the VC that was generated by RAPTOR X was used as input files for prediction. The complex interaction was also analyzed using ClusPro protein–protein docking server (https://cluspro.bu.edu/home.php, accessed on 21 September 2021) [53]. Similar to HDOCK, the PDB of the receptor was 3FXI, and the PDB file of the VC that was generated by RAPTOR X was used as the ligand. The model with the lowest binding energy was chosen and the interacting residues were visualized using PDBsum (http://www.ebi.ac.uk/thornton-srv/databases/pdbsum/Generate.html, accessed on 21 September 2021) [54]. The PROCHECK tool that was integrated into PDBsum calculated the number of residues in the favored region as an indication of a good model.

### 2.19. Molecular Docking of Peptide WSMATYYLF with HLA-A*24:02 and HLA-A*24:07

As a means to validate the T-cell epitope prediction, we docked the peptide to the HLA molecules and analyzed the interaction. There were two steps to perform the analysis. The first step involved determination of HLA structure and generation of the PDB file. PDB structure of HLA-A*24:02 and HLA-A*24:07 were inferred from their residues. The different between HLA-A*24:02 and HLA-A*24:07 was only one residue number 70 in HLA-A*24:02 is histidine, and in HLA-A*24:07 is glutamine. The protein sequences were submitted to I-TASSER server (https://zhanggroup.org/I-TASSER/, accessed on 21 September 2021) to predict the structure. The PDB file of the best protein structure model of HLA-A*24:02 and HLA-A*24:07 was then downloaded from I-Tasser online server after the computation was finished. The best PDB model was defined as the one having the highest C-score. The pdb data were used as the input for the molecular docking analysis. The second step was the molecular docking of peptide WSMATYYLF to the HLA molecule, which was computed through CABS-dock web server (http://biocomp.chem.uw.edu.pl/CABSdock accessed on 26 October 2021) [55,56,57]. The analysis was run with the default CABS-dock mode and the model of the protein–peptide complex with the highest score was considered.

## 3. Results

### 3.1. SARS-CoV-2 ORF1ab Polyprotein Contains Evolutionary Stable Regions with Low Entropy

One important factor that needs to be considered in the vaccine formulation is the conservancy of the epitopes. Therefore, the sequences of ORF1ab from the SARS-CoV-2 Wuhan Hu-1 isolate (NCBI Reference Sequence YP_009724389.1) and its variants including the VOC and VOI were obtained (Table 1) and checked for sequences conservancy.

As T-cells receptors recognize antigen in the form of 9-mer peptide presented by HLA molecule, we checked for the conservancy of the 9-mers using the AVANA tool. The AVANA tool generates Shannon’s entropy, which is a parameter to infer evolutionary stabilities of any given 9-mer sequences within a complete protein. Low entropy values (zero or close to zero) suggest highly conserved positions. The AVANA results showed that the entropy values of SARS-CoV-2 ORF1ab 9-mers range from 0.00 to 1.44. As displayed in Figure 2, the vast majority of 9-mer sequences from ORF1ab were of very low entropy, which suggests that the protein has low variability, high conservancy, and evolutionarily stable. Theoretically, the highest possible 9-mer entropy value is 39 [17]; however, the values are much lower when comparing closely related viral variants. For instance, a similar entropy analysis of SARS-CoV-2 spike protein yielded high occurrence of positions with high entropy (>0.800), identified as mutation hotspots [58].

### 3.2. SARS-CoV-2 ORF1ab Contributes a Large Number of Experimentally Known Immunogenic Epitopes in IEDB

IEDB contains the information about T-cell epitopes that were proven experimentally to be recognized by T-cells in assays such as IFNγ ELISPOT and intracellular cytokine staining flow cytometry. The presence of T-cells recognizing the epitopes indicates that the epitopes are immunogenic. Applying the assumption that the epitopes are mostly 9-mers, the percentage of immunogenic epitopes over the possible number of epitopes generated per protein was calculated (Table 2).

Although only 10% of the possible ORF1ab T-cell epitopes were reported to be immunogenic, ORF1ab contributed significantly (38.5%) to the total number of SARS-CoV-2 T-cell epitopes reported in IEDB. Table 2 shows all proteins of SARS-CoV-2 could potentially be the source of immunogenic T-cell epitopes, as these epitopes were experimentally proven by T-cell assay. A large proportion of epitopes per protein are generated from the spike, membrane, and nucleocapsid protein, with the percentages of 45.4, 59.0, and 44.2%, respectively. Despite only 9.6% of the ORF1ab polyprotein being T-cell epitopes, the number of these immunogenic epitopes contributes 38.4% of the total reported epitopes due to the large size of the protein. A large number of potential epitopes would allow more flexibility in finding peptides that are immunogenic and conserved among SARS-CoV-2 variants. Therefore, peptides from SARS-CoV-2 ORF1ab polyprotein would be useful in the pan-universal SARS-CoV-2 vaccines development.

### 3.3. HLA Allele Frequencies of the Indonesian, Thai, and German Population

In order to see the diversity of HLA alleles in different populations, the HLA alleles of the Indonesian, Thai, and German population and their frequencies were plotted in Figure 3. The most predominant HLA Class I alleles in Indonesian population were HLA-A*24:07 (20.7%), HLA-A*33:03 (16.9%), HLA-A*11:01 (16.4%), HLA-A*24:02 (14.4%), HLA-B*15:13 (11.0%), and HLA-B*15:02 (10.7%), while the most predominant HLA Class II alleles were HLA-DRB1*12:02 (36.8%), HLA-DRB1*15:02 (24.1%), and HLA-DRB1*07:01 (13.7%). Comparing the allele frequency of Indonesia with Thailand and Germany (Figure 3) clearly shows that allele frequency is characteristic for each population. While HLA-A*24:07 had the highest frequency in the Indonesian population (21%), it is much lower in Thailand (5%), and almost none in Germany (0.03%). On the other hand, HLA-A*02:01 was predominant in Germany (27%), but much lower in both Indonesia (7.5%) and Thailand (1.8%). The differences in the allele frequency between populations need to be considered to avoid bias in the formulation of T-cell epitope-based vaccine.

### 3.4. Asian HLA Alleles Are Less Studied as Compared to the HLA Alleles Predominant in the European Population

Table 3 shows the number of SARS-CoV-2 T-cell epitopes that are associated with HLA alleles of the Indonesian, Thai, and German populations. Comparing the number of reported T-cell epitopes presented by HLA alleles revealed that there was no information about epitopes associated with some of the HLA alleles significant for the Asian population such as HLA-A*24:07, A*33:03, and B*15:13. This indicates that these HLA alleles are less studied as compared to the HLA alleles predominant in the European population and stresses the need for more T-cell assays conducted using samples from convalescent individuals from Indonesia and Thailand.

### 3.5. Prediction of CTL Epitopes and Evaluation of Immunogenicity

CTL epitopes from ORF1ab were predicted using NetCTLpan 1.1 against a panel of 56 HLA Class I alleles as shown in Figure 2. NetCTLpan 1.1 analysis generated, in total, 1132 9-mer peptides with the percentile rank less than 1% for the HLA Class I allele. The number of peptides that bind per HLA allele is shown in Figure 4. HLA-A*29:01 (allele frequency of 0.008) binds to the highest number of peptides (126), while HLA-B*13:01 (allele frequency of 0.015) binds the lowest number of peptides (19). The top five most predominant HLA Class I alleles in the Indonesian population HLA-A*24:07 (allele frequency of 0.207), HLA-A*33:03 (0.169), HLA-A*11:01 (0.164), HLA-A*24:02 (0.144), and HLA-B*15:13 (0.11) bind 83, 72, 111, 82, and 71 peptides, respectively. Up to this stage, we were able to identify in silico the peptides that could potentially bind to the understudied HLA alleles (no data in IEDB, as shown in Table 3) such as HLA-A*24:07, A*33:03, and B*15:13.

A total of 1132 peptides were shortlisted further. Peptides bound to only one allele of HLA Class I were removed from the list since they were not preferable due to the possibility of having low population coverage if these peptides were used in a VC. The selected peptides were then evaluated using the IEDB Immunogenicity analysis tool and revealed that 410 peptides were shown to have a positive immunogenicity score. The 9-mer peptides were further screened for the possibility to bind to at least one HLA Class II allele, because binding to both Class I and Class II could be beneficial to invoke more robust immune responses. In the end, only 65 peptides fulfilled the criteria (Table 4) and were therefore selected for further evaluation.

Looking into detail about the immunogenicity score of the 65 peptides, the lowest score was 0.0048 (^6978^YKLMGHFAW^6986^) and the highest was 0.3348 (^6714^FELEDFIPM^6722^). The most immunogenic peptide was already reported in IEDB as binder for HLA-B*40:01. In our prediction, ^6714^FELEDFIPM^6722^ binds to 13 HLA Class I alleles (HLA-B*13:01, HLA-B*15:10, HLA-B*18:01, HLA-B*18:02, HLA-B*37:01, HLA-B*38:02, HLA-B*40:01, HLA-B*40:02, HLA-B*40:06, HLA-B*41:01, HLA-B*44:03, and HLA-B*48:01). The number of HLA alleles that bind to the peptides range from 2 (^6978^YKLMGHFAW^6986^ binds to HLA-B*18:01 and HLA-B*18:02) to 24 (^899^WSMATYYLF^907^ binds to HLA-A*01:01, HLA-A*24:02, HLA-A*24:07, HLA-A*24:10, HLA-A*29:01, HLA-A*32:01, HLA-B*13:01, HLA-B*15:02, HLA-B*15:12, HLA-B*15:13, HLA-B*15:17, HLA-B*15:21, HLA-B*15:25, HLA-B*15:32, HLA-B*18:01, HLA-B*18:02, HLA-B*35:01, HLA-B*35:05, HLA-B*35:30, HLA-B*52:01, HLA-B*56:07, HLA-B*57:01, HLA-B*58:01, and HLA-B*46:01). ^899^WSMATYYLF^907^, the most promiscuous peptide in our list, had also been reported in IEDB as an HLA binder, in particular to HLA-A*24:02.

### 3.6. Prediction of HTL Epitopes and Evaluation of IFNγ Induction Capability

HTL epitopes from ORF1ab were predicted using netHLAIIpan 4.0 against a panel of 22 HLA Class II alleles as shown in Figure 3. As HLA Class II can accommodate longer peptides, the prediction was made for 15-mer peptides. The server generated 792 15-mer peptides as strong binders (≤1% percentile rank) for HLA Class II as shown in Figure 5. HLA-DRB1*15:02 (allele frequency of 0.2410), binds to the highest number of peptides (129), while HLA-DRB1*04:03 and DRB1*04:06 (allele frequency of 0.021 and 0.005, respectively) bind the lowest number of peptides (52). The top three most predominant HLA Class II alleles in the Indonesian population were HLA-DRB1*12:02 (allele frequency of 0.3680), HLA-DRB1*15:02 (0.2410), and HLA-DRB1*07:01 (0.1370). HLA-DRB1*12:02 binds 88 peptides and DRB1*07:01 binds 101 peptides.

Peptide promiscuity (bind to many HLA alleles) is an essential criterion to be fulfilled for a successful vaccine design that can cover as large a population as possible. Out of 792 15-mer peptides, 102 of them bind to at least 5 alleles of HLA Class II and therefore were selected for further evaluation. Production of IFNγ by HTL is important for generation and differentiation of the CD8^+^ T-cell into a cell that has a full effector function, and for the induction of T-cell memory. Out of 102 peptides that were analyzed by the IFNγ prediction server, only 40 peptides have positive scores and are therefore short-listed for further downstream analysis (Table 5).

### 3.7. Conservancy Analysis

Epitopes conservancy among SARS-CoV-2 variants was the next criteria applied to the predicted 65 CTL and 40 HTL epitopes. The conservancy analysis was carried out using the IEDB epitope conservancy analysis tool against ORF1ab sequences from SARS-CoV-2 variants listed in Table 1. The duplicated sequences were removed before the analysis to avoid bias. The goal of the conservancy analysis was to identify epitopes having conservancy levels near to 100% across SARS-CoV-2 variants to be included in the VC.

IEDB T-cell epitope conservancy analysis revealed that the majority of the HTL and CTL epitopes were conserved. Out of 40 HTL epitopes, 26 of them had at least 95% conservancy within the variant and among different variants (Appendix A). These 26 peptides were short-listed for downstream analysis. CTL peptides were conserved within each variant and across all variants, with the level of conservancy mostly above 97% (Appendix A).

One of the exceptions is peptide ^3137^FWITIAYII^3145^ (binds to HLA-A*24:02, HLA-A*24:07, and HLA-A*24:10), which had heterogeneity in its sequence due to mutation F3137S. ^3137^FWITIAYII^3145^ was present only in 9.61% of the ORF1ab sequences of the delta variant B.1.617.2, while the rest of the sequences were ^3137^SWITIAYII^3145^. The changes in amino acid F3137S, however, did not result in the abrogation of the binding of the peptide into HLA molecules; instead, our analysis showed that mutant peptide had a stronger binding affinity (Table 6) toward HLA molecules. The peptide ^3137^FWITIAYII^3145^ is part of the NSP4 (size 500 amino acids) protein of SARS-CoV-2, which is a membrane protein that contains four transmembrane domains. The ^375^FWITIAYII^383^ peptide is located in the fourth transmembrane region of NSP4 [59]. Together with NSP3 and NSP6, NSP4 forms double-membrane vesicles that are needed for viral replication and transcription as well as for protecting the viral RNA from innate immune recognition [59,60,61]. NSP4 demonstrated high conservancy among the other coronaviruses, indicating its importance for viral replication, and during the pandemic of 2020 only one mutation (M324) was detected [16]. However, as the delta variant emerged and diverged into several clades, more mutations in NSP4 were detected including the F375S mutation, which is specific for clade E of the delta variant [62]. Comparing the binding affinity of the ancestral peptide versus the mutant peptide, one might argue that this mutation might not be beneficial for the virus as it can be easily recognized by T-cells. However, this would only happen if the population had the correct HLA alleles. Thus, the F3137S mutation might benefit viral replication, but since it occurred in clade E, it is assumed that the variant delta subclade E occurred in populations lacking HLA-A*24:02, A*24:07, and A*24:10.

### 3.8. Comparison of Predicted Epitopes and Experimentally Proven Epitopes from IEDB

As a means to partially validate the prediction, the in silico identified epitopes were compared with the experimentally proven epitopes that are curated in IEDB. Out of 65 predicted CTL epitopes (9-mer), 26 matched with experimentally proven 9-mer epitopes by T-cell assay, 20 by HLA assay, and 10 by both T-cell and HLA assays (Table 4). Many of the experimentally proven epitopes are presented by only one HLA allele, such as HLA-A*02:01 or A*24:02. However, in our in-silico analysis, the peptides were predicted to bind to many other HLA alleles with strong binding affinity. Out of 40 predicted 15-mer HTL epitopes, 4 (^5776^VSALVYDNKLKAHKD^5790^, ^5019^PNMLRIMASLVLARK^5033^, ^4561^PDILRVYANLGERVR^4575^, and ^1350^KSAFYILPSIISNEK^1364^) matched with experimentally proven 15-mer epitopes by T-cell assay. All four HTL epitopes were reported in IEDB as positive for IFNγ ELISPOT assay, which confirmed the IFNγ prediction that was conducted in our in silico study.

### 3.9. Homology with Human Peptides

Homology of 40 HTL epitopes (15-mer peptides) with human peptides was conducted by analyzing the 9-mer peptide core component. That is because 9-mer (residue 4–12 of the 15-mer) is the core component of the peptide whose side chains interact with the HLA molecule and interact with the T-cell receptor. BlastP analysis showed that none of the HTL 9-mers were homologous with the 9-mer from the human protein. Therefore, we did not analyze the HTL 9-mers further but focused on the CTL 9-mer peptides, only.

Homology analysis of the 65 CTL epitopes revealed that none have 100% homologies (9/9) with human peptides. However, one epitope has eight contiguous amino acids that matched 100% with the human peptides and six epitopes have seven contiguous amino acids that matched with the human peptides as shown in Table 7. NetCTLpan analysis of these human peptides showed that these self-peptides were predicted as T-cell epitopes with similar binding affinity to the HLA molecules that present the corresponding SARS-CoV-2 peptide (Table 7 compare with Table 4). SARS-CoV-2 peptides having high similarity to several human proteins have also been reported by other [63].

Interesting to note is the ORF1ab peptide ^2784^AIFYLITPV^2792^ matched with human peptide AIFYLITLV, which is derived from the olfactory receptor (EAX03180.1). Human peptide AIFYLITLV was predicted to be presented by HLA-A*02:01, HLA-A*02:03, HLA-A*02:06, similarly to the SARS-CoV-2 peptide counterpart. It is possible that high homology will result in the activation of cross-reactive T-cells recognizing SARS-CoV-2 peptides to attack olfactory cells and cause of anosmia in some COVID-19 patients. The recent publication confirmed experimentally that epitope ^2784^AIFYLITPV^2792^ was recognized by T-cells from SARS-CoV-2 convalescent individuals having HLA-A*02:01 allotype [64]. It would be interesting to check the patient’s history of anosmia symptoms and whether the symptoms disappeared or persisted, due to the activation of self-reactive T-cells.

### 3.10. Epitope Cross-Reactivity with Human Peptides, Human Common Cold Coronaviruses (HCCs), or Other Ubiquitous Antigens

As shown in Table 7, seven SARS-CoV-2 ORF1ab peptides shared sequence similarities with human peptides. Cross-checking with the IEDB data, it was revealed that six out of seven SARS-CoV-2 ORF1ab peptides that were similar to human peptides were already experimentally confirmed and reported in IEDB (Table 7). One peptide was experimentally confirmed by HLA binding assay, two peptides were confirmed by T-cell assay, and three were confirmed by both T-cell and HLA binding assay. Here, we focused our analysis on the epitopes that had been confirmed by T-cell assay and checked whether or not the epitopes were recognized by SARS-CoV-2 convalescent individuals or healthy subjects who never experienced SARS-CoV-2 infection. Four ORF1ab peptides (^5614^FAIGLALYY^5622^, ^3684^YASAVVLLI^3692^, ^6748^LLLDDFVEI^6756^, and ^3752^FLARGIVFM^3760^) that matched with human peptides were recognized by T-cell from healthy individuals who have not been infected by SARS-CoV-2 (Appendix A). As a comparison (Appendix A), from 23 IEDB ORF1ab peptides that did not match with human peptides, only 7 were confirmed by T-cell assay using samples from healthy individuals who never experienced SARS-CoV-2 infection while the rest were detected in individuals previously infected by SARS-CoV-2. Several studies reported the presence of cross-reactive T-cells recognizing epitopes from SARS-CoV-2 in individuals that were never exposed to SARS-CoV-2 [25,65,66,67,68,69]. We then checked for the degree of homology between SARS-CoV-2 peptides versus human peptides and HCCs. In some instances, SARS-CoV-2 peptides shared greater homology with human peptides rather than with HCC peptides (Appendix A).

The fact that the epitopes are similar to human self-peptides and the T-cells recognizing these peptides were mostly found in the healthy individuals without prior exposure to SARS-CoV-2, raises three possibilities. The first possibility is that T-cells responding to these peptides were primed by exposure to HCCs. The second possibility is that the assay picked up the signal and detected the presence of self-reactive T-cells in the circulation. Although the presence of highly self-reactive T-cells in the circulation is highly unlikely due to negative selection in the thymus, positive selection will permit T-cells to be slightly reactive toward self-antigens. One study reported that SARS-CoV-2 proteomes contain peptides similar to human proteomes and might able to trigger autoimmunity [70].

Appendix A shows SARS-CoV-2 peptides that did not have homology with human peptides. Some of these SARS-CoV-2 peptides (i.e., ^1674^YLATALLTL^1682^, ^2787^YLITPVHVM^2795^, ^2786^FYLITPVHV^2794^, and ^3121^FLAHIQWMV^3129^,) also did not have homology with HCC. Interestingly, these peptides were recognized by healthy individuals. These data suggest the third possibility by which the T-cells, recognizing the epitopes, are primed by the exposure to other ubiquitous antigens. A recent report suggested that such sequence homology exists between SARS-CoV-2 peptides and peptides from allergen proteins [71], malaria proteins [72], and antigenic proteins in BCG, OPV, MMR, and some other vaccines [73]. Perhaps sequence homology, in the context of 9-mer T-cell epitopes, between pathogens is more common than what was originally thought.

Sequence homology between SARS-CoV-2 peptides and HCC peptides (Appendix A) strongly supports the hypothesis that T-cells primed by previous HCC infection can recognize SARS-CoV-2. Whether having such cross-reactive T-cells in the circulation is beneficial and leads to better disease outcomes, or detrimental and leads to severe disease, is yet to be determined. In some individuals, the previous infection with HCC might protect them from severe disease, or even lead to asymptomatic infection. Our sequence analysis and cross-checking with IEDB data showed that peptide ^2883^FLPRVFSAV^2891^, which shares homology with OC43 FLRVVFSQV, was recognized by an individual with documented exposure to SARS-CoV2 but without evidence for disease (Appendix A) [74]. Although the evidence that the individual had prior OC43 infection remains to be established, it suggested the potential protection from pre-existing T-cells primed by HCC infection.

### 3.11. Epitope Selection

Based on the criteria, such as highest percentile rank (<1%) in the prediction, epitopes promiscuity, immunogenicity, IFNγ induction ability, high conservancy across all variants, low entropy value, and the absence of homology with human peptides, seven CTL and five HTL epitopes were chosen to be included in the VC (Table 8). The epitopes were chosen so that a minimum number of epitopes could cover the largest population possible (accommodate all HLA alleles in the population).

### 3.12. Population Coverage

Multi-epitope peptide-based vaccines that induce cell-mediated immunity need to be constructed from promiscuous epitopes to cover a large population since T-cells recognize antigen in the form of peptide complex with HLA molecules, and HLA is the most polymorphic genes in humans with allele frequency varying by ethnic groups. We used the IEDB population coverage analysis tool to calculate the coverage of each chosen epitope and the epitope set for Indonesia, Thailand, Germany, and the world population as shown in Table 8. Table 9 shows the population coverage for 12 chosen T-cell epitopes presented by HLA class I, class II, and class combined. The majority of the epitopes would be responded by the Indonesian people, which would recognize 8–9 epitopes hits/HLA combinations and 90% of the population would recognize a minimum of 6 epitopes/HLA combinations. A combination of these 12 chosen epitopes was shown to have good coverage not only for Indonesia (100%) but also for Thailand (100%), Germany (99.98%), and the world (99.88%). Hence these 12 epitopes were chosen as candidates for vaccine design.

### 3.13. Vaccine Design 

The vaccine (Figure 6) was constructed by combining five HTL and seven CTL epitopes using linkers such as GPGPG and AAG, respectively [36]. Linkers were used to facilitate the antigen processing inside the cells and to ensure that individual epitopes will be generated by the cell. β-defensin was incorporated using EAAAK linker at the N-terminal to increase the antigenicity and immunogenicity of the peptide-based vaccine. β-defensin will also act as a TLR4 ligand that will induce the maturation of antigen-presenting cells and the successful activation of T-cells in the lymph nodes.

### 3.14. Vaccine Antigenicity, Allergenicity, Toxicity, and Physicochemical Characteristics

The VC was predicted to be a probable antigen as the antigenicity score was calculated to be 0.4369 by VaxiJen 2.0 and non-allergenic as predicted by AllergenFP v.1.0. Expasy ProtParam tool calculated the physicochemical properties of the VC, composing of 212 amino acids and a molecular weight of 22.993 kDa. The theoretical pI was 9.39, which indicated that the VC is slightly basic. The estimated half-life of the VC in *Escherichia coli* in vivo, yeast in vivo, and mammalian red blood cells in vitro, is 10, 20, and 30 h, respectively. This indicated that the VC could be synthesized using these cell systems. The VC was predicted to be stable as the instability index was computed to be 32.23, thermostable as indicated by the aliphatic index of 84.34, and slightly hydrophobic as the GRAVY score was computed to be 0.065. The VC was evaluated for toxicity using Toxinpred that generated fragments of 10 amino acid lengths and predicted their toxicity (Appendix A). All CTL and HTL epitopes components of the vaccine were non-toxic. Toxic peptides were predicted from the β-defensin part (residue 29–48), which is expected given the fact that β-defensin acts as an adjuvant.

### 3.15. Re-Analyze the VC for Epitopes Generation and Homology with Human Proteins and Microbiomes

The VC was re-analyzed using NetCTLpan1.1 and NetMHCIIpan4.0 to check that the CTL and HTL epitopes used to design the vaccine will be processed and generated by the antigen-presenting cells. Further analysis was conducted to check that the new CTL and HTL epitopes that were generated did not have similarities with the human peptides and peptides from human microbiomes, which will induce autoimmunity, reduce vaccine immunogenicity, and disrupt immune homeostasis.

NetCTLpan1.1. analysis of the VC against 56 HLA Class I alleles revealed that 43 of 9-mer CTL epitopes can be generated from the vaccine (Appendix A). All seven CTL epitopes that were used to construct the vaccine were generated, albeit not all of them were presented by the HLA alleles that were initially predicted to bind. However, this is mainly due to the protein size rather than the changes in binding affinity. The smaller the protein size, the lower the possibility that the epitopes will have a score <1% percentile rank. BlastP analysis against the human proteome showed some of the new epitopes are homologous (100% match) to human self-peptides; in particular, the epitopes from the β-defensin region. The other epitopes are only partially homologous (7/7 amino acid).

NetMHCIIpan4.0 analysis of the VC (Appendix A) showed that all five HTL epitopes were generated (marked with *) along with extra new epitopes. The 9-mer peptide-binding core of all HTL epitopes was evaluated for sequence similarity with the human peptides. None of the HTL 9-mer peptides were homologous (100% match) with human peptides, and only two 9-mer peptides were partially homologous (7/7 amino acid) (marked with ** in Appendix A). Peptide FVSLAAGFE contains residue that matches with heptamer VSLAAGF from human protein hCG2019424 (sequence ID EAX10398.1). Peptide VVISSDGPG contains residues that match with heptamer VVISSDG from guanine nucleotide-binding protein (G protein) (sequence ID EAW53700.1).

None of the CTL and HTL 9-mers have sequence similarities with the peptides from human microbiomes as revealed by the analysis using PBIT (Appendix A). It means that the vaccine construct should not disrupt host immune homeostasis.

### 3.16. In Silico Immune Simulation of the VC

The VC was analyzed using CimmSim for the ability to generate cell-mediated immunity CD8^+^ CTL and CD4^+^ HTL. As shown in Figure 7A, the level of the HTL population increased to around 4000 (cells/mm^3^) after the first dose of vaccine was administered. The number of HTL increased to 10,200 after the second dose of vaccine was given, and 9800 after the third dose. The increase in the number of HTL was accompanied by an increase in the level of HTL memory cells, where the level remains high at 600 after 300 days. The HTLs induced by the vaccine are all in the active, duplicating, and resting state, with no formation of an anergic state (Figure 7B). The absence of anergic T-cells is a good sign that the VC provides enough signal for the TLR and other PRR to sufficiently activate and induce the maturation of dendritic cells.

The level of CTL increased up to 1150 cells/mm^3^ upon administration of vaccine’s first dose (Figure 7C) and then fluctuated between 1055 cells/mm^3^ at the lowest and 1130 cells/mm^3^ at the highest. Administration of the first dose of vaccine increased the level of active state CTL to 1050 cells/mm^3^, which plateaued for 100 days before eventually declining and shifting into the resting state (Figure 7D). The fact that all CTL were found as active, duplicating, and resting-state cells, with no detectable level of anergic cells, indicated that the vaccine can induce dendritic cells maturation and expression of costimulatory molecules, which are needed for successful T-cell activation.

The vaccine did not induce any changes in the number of NK cells population as shown in Figure 7E where the number of NK cells fluctuated between 310 and 380 cells/mm^3^. Each vaccine dose administration induced an increase in the number of dendritic cells that internalize and present the antigen on both HLA Class I and II (Figure 7F). However, the vaccine did not increase the number of active dendritic cells as the population remains constant at 25 cells/mm^3^ throughout the simulated period (350 days). Aside from dendritic cells, the number of macrophages that internalized and presented the antigen on HLA Class II also increased at each dose of vaccine administration (Figure 7G). The number of active and resting macrophages increased concomitantly until the antigens are no longer be detected; at which time, the number of active macrophages declined, and resting macrophages increased.

At each vaccine dose administration, the level of cytokines, notably IFNγ, is significant (410,000 ng/mL, 400,000 ng/mL, and 375,000 ng/mL at the first, second, and third dose, respectively) (Figure 7H). The production of IFNγ confirmed that the selected epitopes in the VC were able to induce Th1 cells, which are needed for the immune responses against viral infection. IL-2 was also produced as a response to the vaccine dose administration. IL-2 is an autocrine signaling protein produced by activated T-cell and acts as a T-cell growth factor. The presence of IL-2 is a good indication that the vaccine will induce T-cell clonal expansion.

### 3.17. Secondary Structure and Tertiary Structure of Vaccine Construct

The secondary structure of the VC was predicted based on its amino acid sequences (Figure 8A) by SOPMA and the tertiary structure was predicted by RAPTOR X. SOPMA server predicts the secondary structure based on the multiple alignments of protein sequences of known structures. The VC had 212 residues of which mainly 95 residues were predicted to adopt the α-helix (44.81%), followed by 59 residues as a random coil (27.83%), 47 residues as an extended strand (22.17%), and only 11 residues were observed in β-turn (5.19%) (Figure 8B). The location and propensity of the secondary structure are shown in Figure 8C,D. A different method was employed to predict the tertiary structure of the VC (Figure 8E). RAPTOR X employed a deep learning method to predict the tertiary structure based on the inter-residue distance distribution of a protein, which worked best for predicting a protein structure that does not have many sequences homology. Overall model quality, indicated by z-score, was generated by ProSA-web, which is a web-based protein structure analysis. Since the size of the VC is 212 amino acids, the z-score should fall between −1 and −10, according to the plot. The z-score of the VC is −7.25, which fell within the range of conformational parameters of native proteins.

### 3.18. Molecular Docking of the VC with TLR4

The association of the antigen molecule with the immune receptor is an essential step for the appropriate activation of the immune responses. Toll-like receptor (TLR) is a pathogen recognition receptor on the surface of the immune cells such as dendritic cells that are important for their respective maturation process. Upon maturation, dendritic cells will be able to migrate from the tissue to the lymph nodes and present the peptide-HLA complex along with costimulatory molecules to activate naïve T-cells. Therefore, molecular docking analysis was performed to analyze the interaction between TLR4 and the VC. HDOCK was employed to generate information about the interacting residues on TLR4 and the VC. The PDB files of TLR4 (3FXI) and the VC generated by RAPTOR X was used as input files for prediction. HDOCK docking simulation generated several models with the top 10 models are listed in Appendix A. The best model was model 1 with a docking score of −283.87, which reflected good interactions between the VCs and TLR4 (Appendix A).

The interaction between the VC and TLR4 was also simulated using ClusPRO by generating a PDB file of the complex for both the TLR (receptor) and the vaccine (ligand). The input files for ClusPRO were the PDB ID of the TLR4 (3FXI) and the PDB file of the VC, which was generated previously by RAPTOR X. The PDB model of the complex between TLR4 and the VC generated from ClusPRO was then used as input for the PDBsum to generate the model and calculate the protein–protein interaction parameters. The model of the complex between TLR4 and the VC is shown in Figure 9. TLR4 is shown in the purple color, while the VC is shown in the red color, as shown in Figure 9A. The complex was formed via the interaction of 27 residues of TLR4 and 24 residues of the VC (Figure 9B). As expected, 16 out of 24 residues in the VC were from the β-defensin, as TLR4 ligand (Figure 9C). The interface area was 1192 Å2 and 1233 Å2 for TLR4 and the VC, respectively. It was observed that the interaction between residues was composed of 12 hydrogen bonds and 8 salt bridges, indicating good docking interaction, which corroborated the docking score generated previously by HDOCK (Appendix A). Ramachandran plot and Procheck results for the complex between TLR4 and the VC showed 70.9% amino acids were in the most favored region and 26.7% in the additional allowed region, 0.5% in the generously allowed region, and only 1.8% in a disallowed region (Figure 9D). Even though less than 90% amino acids were in the most favored region, it is predicted that the complex between the VC and TLR4 would still be generated and that immune responses to the VC will still be induced, as shown by the CimmSim analysis results.

### 3.19. Molecular Docking Simulation of Peptide Binding to HLA-A*24:02 and HLA-A*24:07

WSMATYYLF has been reported in IEDB [24] and experimentally proven as an HLA-A*24:02 binder [75], and immunogenic, in the T-cell assay [64]. However, in this study, we predicted that WSMATYYLF is highly promiscuous and has high population coverage as shown in Table 8. WSMATYYLF binds to 23 other HLA Class I alleles including HLA-A*24:07. Since HLA-A*24:07 is very important for Indonesian and other Southeast Asia populations, we conducted a molecular docking analysis to confirm that peptide WSMATYYLF will bind to HLA-A*24:07. The results show that in principle, WSMATYYLF binds equally well to both HLA-A*24:02 and HLA-A*24:07 (Figure 10A,B), albeit with a slightly different binding mode as reflected in the amino acid residues involved in the binding (Figure 10C,D).

## 4. Discussion

Adaptive immune responses, both humoral and cellular are important to combat viral infection. Humoral immunity, mediated by antibodies produced by B-cells will bind to the virus and hence prevent virus entry into the cells. Cellular immunity mediated by T-cells will kill the infected cells, and hence remove the viral reservoir so that the infection to other cells will be prevented.

In SARS-CoV-2 infection, however, available reports generally suggest a rapid decrease in the SARS-CoV-2-specific antibodies [76,77,78,79]. Confirmed by a longitudinal study showing the level of neutralizing antibodies declines over time after infection [80]. Moreover, a high virus mutation rate resulting in the emergence of SARS-CoV-2 variants, which have different antigenic profiles compared to the ancestral virus, could lead to viral escape from neutralizing antibodies elicited previously by vaccination (vaccine breakthrough) or natural infection (re-infection) [81]. On the other hand, T-cell responses could last up to 10 months after infection [82]. On top of that, intriguing data shows that host protection against COVID-19 could be mediated solely by T-cells. This is evident in some COVID-19 patients who have hematological malignancy comorbidity and therefore need to receive anti-CD20 therapy. Anti-CD20 therapy is part of the treatment for hematological malignancy [83] and autoimmune disorders like multiple sclerosis [84], which results in the depletion of B-cells, and hence these patients can not mount an antibody response. However, despite the lower level of IgG in these patients, SARS-CoV-2 specific T-cell responses were detected, and the level was associated with good clinical outcomes [85]. The data suggests that T-cells play a significant role in protection in the situation where the neutralizing antibody is not present.

Preexisting T-cell immunity could ameliorate progression to severe COVID-19 as the early and robust T-cell responses to SARS-CoV-2 have been associated with less severe diseases [86]. In addition, patients with mild COVID-19 have been shown to have enriched CD8^+^ T cells specific for conserved epitopes across HCCs [87]. Altogether, these advocate for a T-cell oriented strategy for COVID-19 vaccines [88]. Knowledge and experiences from the other two zoonotic coronaviruses (CoV)-SARS-CoV-1 and MERS-CoV have confirmed the importance of T-cell immunity in the recovery and long-term protection from coronavirus infections [89,90,91]. Moreover, data from studies on humoral immunity to SARS-CoV-1 demonstrated that antibody responses were short-lived, whereas memory T-cell responses were long-lasting that could be detected at least 17 years after infection [68].

T-cells recognize peptides, derived from the pathogen, which are presented as a complex with the HLA molecule. The peptide, termed epitope, is usually 8–10 amino acids long for presentation by HLA Class I, and 15–20 amino acid long for HLA Class II. Each of the peptide–HLA Class I and –Class II complex is recognized by CD8^+^ and CD4^+^ T-cells, respectively. The HLA molecule, also known as HLA (human leukocyte antigen) in humans, is polymorphic. This polymorphism resulted in many different HLA alleles existing within the human population [92] and each population can be characterized by the different frequencies in the HLA allotypes. As an example, HLA-A*02:01 is predominant in the Caucasian population with allele frequency up to 40%, but only around 6% in the Indonesian population. On the other hand, HLA-A*24:07 is very common in the Indonesian population with an allele frequency of around 26%, but less than 1% in the Caucasian population. As the HLA molecule presents peptide antigen to T-cells, the type of peptide and strength of peptide–HLA association differs among distinct populations, as well as among racial and/or ethnic groups. Thus, each population might have preferences toward specific peptide epitopes to invoke T-cell mediated immunity.

Several CD4^+^ and CD8^+^ T-cell epitopes have been identified for SARS-CoV-2 and the data are curated in IEDB. Most of the identified T-cell epitopes were limited to several HLA allotypes that are predominant in the Caucasian population, such as HLA-A*02:01, whereas there were no data about peptides that were presented by HLA-A*24:07. Therefore, in this study, we aimed to identify the most promiscuous T-cell epitopes from SARS-CoV-2 to be used in the vaccine formulation for the world population while still considering the HLA alleles predominant in Indonesia that are not yet well studied. An in silico study identifying T-cell epitopes presented in the South American population has also been conducted by reviewing the HLA alleles frequencies in those countries [93].

There have been several reports employing immunoinformatics to identify T-cell epitopes and formulate a vaccine for SARS-CoV-2 infection. PubMed search using keywords ‘peptide-based vaccine’, ‘SARS-CoV-2’, and ‘immunoinformatics’ conducted on 14 October 2021 resulted in 15 research articles [94,95,96,97,98,99,100,101,102,103,104,105,106,107,108]. In this study, we made a prediction based on the HLA alleles that are present with at least 5% frequency in the Indonesian population. The majority of these HLA alleles have not been well studied, not only for the SARS-CoV-2 T-cell epitopes but also for other infectious agents, and hence no experimental data were available. In this analysis, we included some HLA alleles of the Thai population as well as HLA alleles included in the study of SARS-CoV-2 T-cell epitopes conducted in Germany [25]. In total, 56 HLA Class I and 22 HLA Class II alleles were covered in this study. Given that one individual can have 6 types of HLA class I and 2 types of HLA DRB1, by selecting these 78 alleles, it is estimated that 99% of people in the world have at least one HLA class I allele and 90% have one HLA Class II allele listed here; therefore, the vaccine construct could cover a large proportion of the world’s population.

We focused our search on promiscuous T-cell epitopes in the ORF1ab polyproteins since they would be beneficial for vaccine population coverage. Several studies support the utilization of ORF1ab as a vaccine target. A study by Gangaev et al. (2021) showed that ORF1ab contains an immunodominant epitope restricted by HLA-A*01:01 and the epitope-specific CD8^+^ T-cellwas detectable up to 5 months after recovery from critical and severe COVID-19 diseases [109]. Other studies [69,110] also confirmed that ORF1ab is the most immunogenic region of SARS-CoV-2 and contains the majority of the highly conserved immunodominant epitopes. Looking at the number of ORF1ab T-cell epitopes in the IEDB data (Table 2), it is evident that ORF1ab contains immunogenic epitopes that could be used for vaccine development.

ORF1ab is quite conserved as shown by entropy analysis. The high conservancy could be due to the function of ORF1ab as replicase enzymes that are needed for the virus to successfully replicate inside the host cells. ORF1ab is also the first protein to be synthesized by the infected cells [10], therefore having T-cells that are primed to recognize epitopes from ORF1ab will be beneficial for early viral clearance. Since ORF1ab contributes the highest numbers of the experimentally confirmed immunodominant epitopes, a detailed evaluation of these T cell epitopes for promiscuity and conservancy, which is important for vaccine design, will thus be possible. Our immunoinformatics analysis using NetCTLpan predicted 1132 CTL peptides and NetMHCIIpan predicted 792 HTL peptides that bind, respectively, to at least 1 HLA Class I and 1 HLA Class II allele with the percentile rank less than 1%, minimizing the false-positive results.

The predicted peptides were evaluated further using several criteria such as immunogenicity, IFNγ inducing ability, promiscuity in binding to HLA alleles, conservancy across all SARS-CoV-2 variants, low entropy value, and non-homology with the human peptides and human microbiome peptides. In the end, seven CTL and five HTL epitopes were chosen to be incorporated in the VC. The VC covers the entire Indonesian and Thai population (100.00%) and a little less than 100.00% for Germany and the entire world population (>99.9%). In this study, the VC was evaluated further and fulfilled criteria, such as good antigenicity, non-allergenicity, and non-toxicity, and had good physicochemical characteristics, such as pI, stability, half-life, and GRAVY score. Similar approaches have already been used by previous studies to obtain peptides as vaccine candidates and for evaluation of vaccine physicochemical properties, as reported in the review by Sohail et al. (2021) [111].

The vaccine should not disrupt immune homeostasis or induce autoimmunity; therefore, in this study, we evaluated the VC for similarity with human peptides and human microbiomes. We used the VC sequences as input for NetCTLpan and NetMHCIIpan and obtained the list of potential peptides to be generated with percentile rank <1%. The results showed that all SARS-CoV-2 CTL and HTL epitopes that were used to construct the vaccines were indeed generated. However, other new extra CTL and HTL epitopes were also generated. These epitopes encompassed the peptide linkers at the junctional region between the original HTL and CTL epitopes. The generation of these extra epitopes has not previously been reported by others. While the generation of these extra epitopes cannot be avoided, we need to check whether the sequence of extra epitopes is homologous with the human peptides or human microbiomes. The BlastP analysis and PBIT analysis of these new CTL and HTL epitopes showed no similarity with the human peptides and human microbiomes, which confirmed the safety profile of the VC.

The safety profile is not the only requirement for a vaccine. The vaccine component should be able to interact with the receptor on the surface of the immune cells to generate appropriate immune responses. The interaction between the vaccine and the TLR4 was evaluated in this study. Engagement of TLR on the surface of dendritic cells with the ligand will ensure the proper maturation of the dendritic cell. Mature dendritic cell will be able to process the vaccine antigen, and then present the peptide–HLA complex to be recognized by T-cells as the signal1. Mature DC will also upregulate the CD80/CD86 molecules that will engage CD28 on the T-cell, which constitutes the required signal 2 for T-cell activation. The molecular docking simulation of the VC and TLR4 revealed that there are sufficient interactions between the two molecules to induce robust immune responses (Figure 9). This corroborated the immune simulation analysis results (Figure 7), which show the generation of CTL and HTL responses upon vaccine administration along with the production of cytokines such as IFNγ and IL-2.

Within our VC, ^899^WSMATYYLF^907^ is the most promiscuous peptide that binds to 24 HLA Class I alleles, covering 94.80%, 77.44%, 66.25%, and 64.13% of Indonesia, Thailand, Germany, and the world population, respectively. ^899^WSMATYYLF^907^ has been reported in IEDB by HLA binding assay to HLA-A*24:02 [75] and T-cell assay using samples from an A*24:02 positive individual [64]. We performed the docking between peptide ^899^WSMATYYLF^907^ with HLA-A*24:07, which is the most predominant allele in the Indonesian population. So far, there are no experimental data of SARS-CoV-2 peptide that is represented by HLA-A*24:07. Despite its importance for the Indonesian and Southeast Asian populations, the HLA-A*24:07 is less studied as compared to HLA-A*24:02. In IEDB, there is only one pathogen peptide reported to be presented by HLA-A*24:07 [112]. Therefore, it is interesting to analyze the interaction between HLA-A*24:07 with the immunogenic peptide of SARS-CoV-2 (^899^WSMATYYLF^907^). From the docking analysis it is shown that the best model of HLA-A*24:07 with peptide ^899^WSMATYYLF^907^ has a more accuracy compared with HLA-A*24:02, suggesting that the peptide can potentially be presented by HLA-A*24:07, thus confirming our CTL epitope prediction.

## 5. Conclusions

T-cells recognize infected cells based on the complex of a pathogenic peptide and a HLA molecule. High polymorphism of the HLA gene ensures that any antigens can be presented to the immune system. However, the difference in HLA allotypes among different populations greatly affects the characters of adaptive immune responses against viral diseases and vaccines. These differences are evident between the Southeast Asian and European populations. Thus, regarding the COVID-19 pandemic, the identification and characterization of conserved SARS-CoV-2 T-cell epitopes across different HLA allotypes provide wide-ranging applications for diagnostic, prophylactic, and therapeutic developments. A vaccine applying conserved SARS-CoV-2 epitopes that induce both memory CD8^+^ and CD4^+^ T-cell responses across different populations with various HLA allotypes might represent a promising tool to end the public health and economic burdens due to the COVID-19 pandemic.

The current study generated data about SARS-CoV-2 ORF1ab T-cell epitopes and their characteristics, such as the epitopes conservancy; HLA-binding promiscuity; and the level of homology with peptides from human common cold coronaviruses, human self-proteins, and microbiomes. Those characteristics are important for the development of a peptide-based vaccine that induce T-cell responses. T-cells target the antigens originating from all proteins of SARS-CoV-2, including ORF1ab. ORF1ab is intrinsically conserved because it is important for virus replication, and therefore not easily mutated. Vaccines based on the evolutionarily stable protein is beneficial because it will work against all variants of SARS-CoV-2. The current study nominated 12 conserved and promiscuous epitopes to be used in the vaccine development that will cover the majority of the Indonesian and the world population. One epitope in particular, ^899^WSMATYYLF^907^, was predicted to bind to HLA-A*24:07, which is the HLA allele predominant in the Indonesian population.

We highlighted Indonesia in this study since the HLA background of the population is different to that of the Caucasian population, as shown in Figure 3. HLA-A*24:07 is not very well studied and no data are available about SARS-CoV-2 T-cell epitopes associated with this HLA. The in silico data generated in this study should be followed by wet-lab experiments to map T-cell epitopes that will be recognized by COVID-19 convalescent individuals from Indonesia. Such a study has not previously been conducted, even though the allele frequency for HLA-A*24:07 is significantly high (0.26) and Indonesian (population of 277 million) is the fourth largest population in the world, and is also affected by the pandemic.

The study also generated other interesting findings related to the cross-reactive epitopes between SARS-CoV-2 and human proteins. Epitope ^2784^AIFYLITPV^2792^ matched with the human peptide AIFYLITLV, which is derived from the olfactory receptor. We hypothesized that epitope similarity might contribute to the anosmia symptoms in some COVID-19 patients. Experimental validation is needed to test the epitopes and the characteristics of T-cells recognizing the epitopes. The data generated might entangle the molecular mechanism of anosmia in some patients.

Taken together, the peptides reported here will provide more insight into the cellular-mediated immune responses against SARS-CoV-2 in populations with different genetic backgrounds and environments that could bring novel ideas for the development of COVID-19 vaccines and immune monitoring, which could be effective across different populations worldwide.

## Figures and Tables

**Figure 1 vaccines-09-01459-f001:**
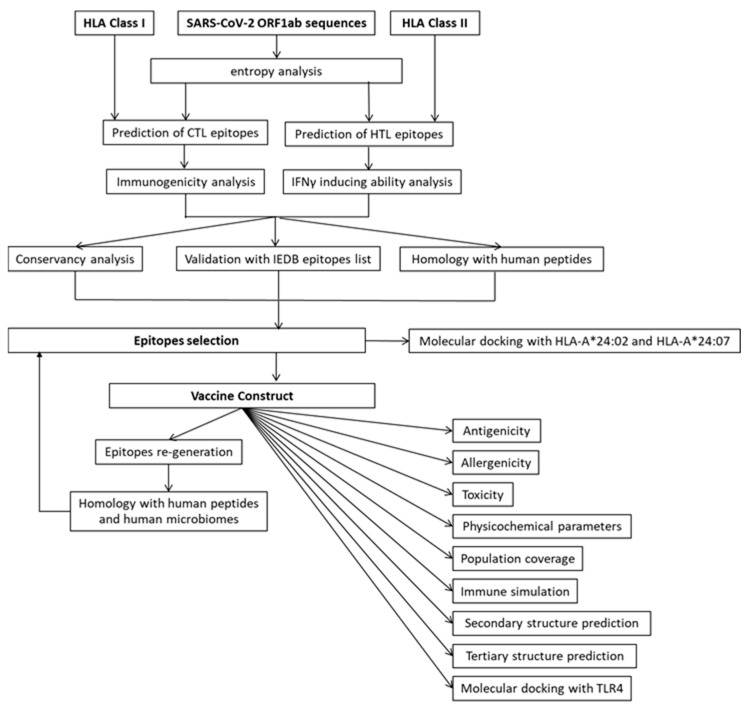
Flowchart indicating the immunoinformatics methods used in this study to identify the CTL and HTL epitopes from SARS-CoV-2 ORF1ab polyprotein.

**Figure 2 vaccines-09-01459-f002:**
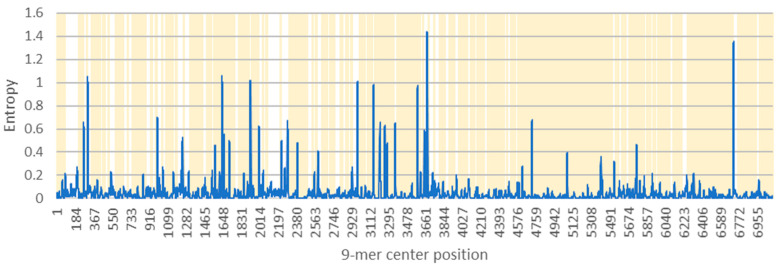
Entropy values of SARS-CoV-2 ORF1ab protein 9-mers. Blue line denotes the entropy values for 9-mer in the corresponding center position. Yellow vertical lines in the background depicts the occurrence of 0 entropy.

**Figure 3 vaccines-09-01459-f003:**
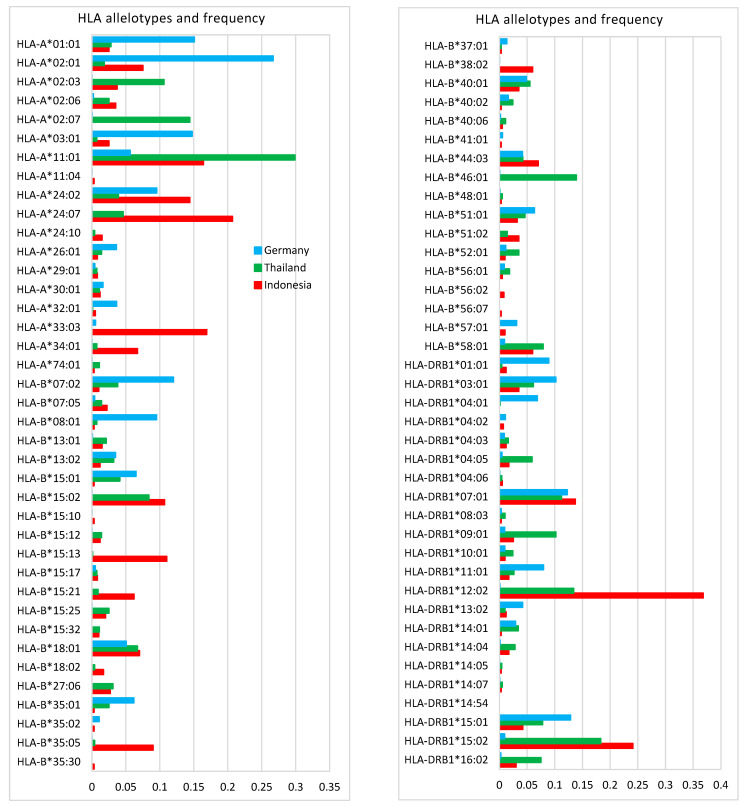
Indonesian HLA alleles and frequency. There were 56 HLA Class I and 22 HLA Class II alleles included in the study. The allele frequency in the Indonesian population (red bar) was compared to those in Thailand (Green bar) and Germany (Blue bar).

**Figure 4 vaccines-09-01459-f004:**
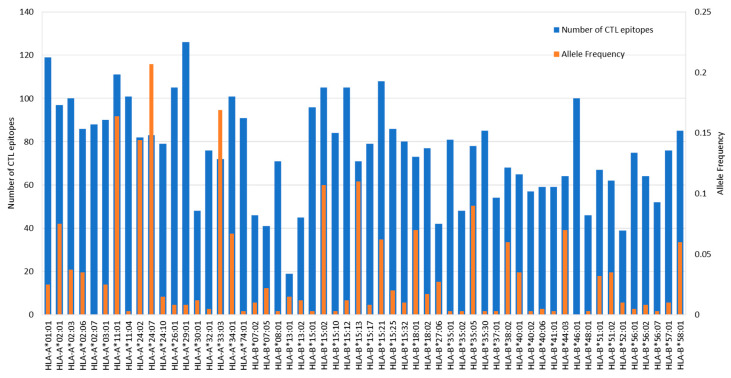
The number of predicted CTL epitopes that are presented by the HLA Class I allele. NetCTLpan 1.1 predicted 1132 9-mer peptides that bind to at least 1 HLA Class I alleles with the percentile rank of less than 1%.

**Figure 5 vaccines-09-01459-f005:**
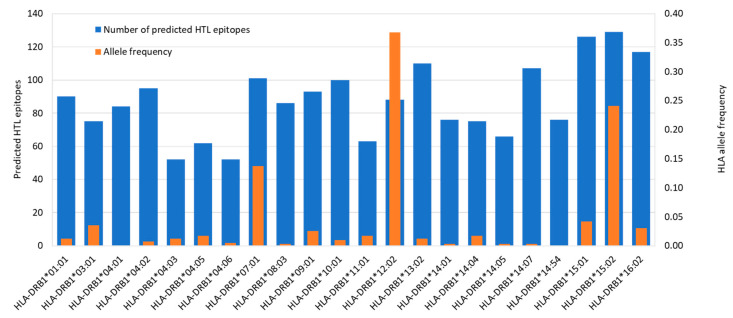
The number of predicted HTL epitopes that are presented by the HLA-DRB1 allele. NetHLAIIpan4.0 predicted 792 peptides bind to at least 1 HLA Class II with a strong binding affinity (≤1% percentile rank).

**Figure 6 vaccines-09-01459-f006:**
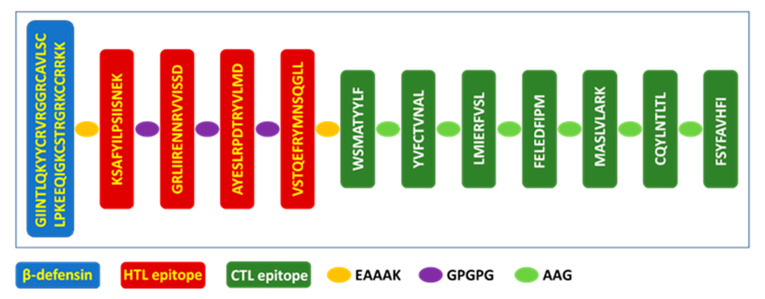
Amino acid sequences of the VC. Beta defensin (was used as adjuvant, HTL epitopes, and CTL epitopes were connected using linkers EAAAK, GPGPG, and AAG.

**Figure 7 vaccines-09-01459-f007:**
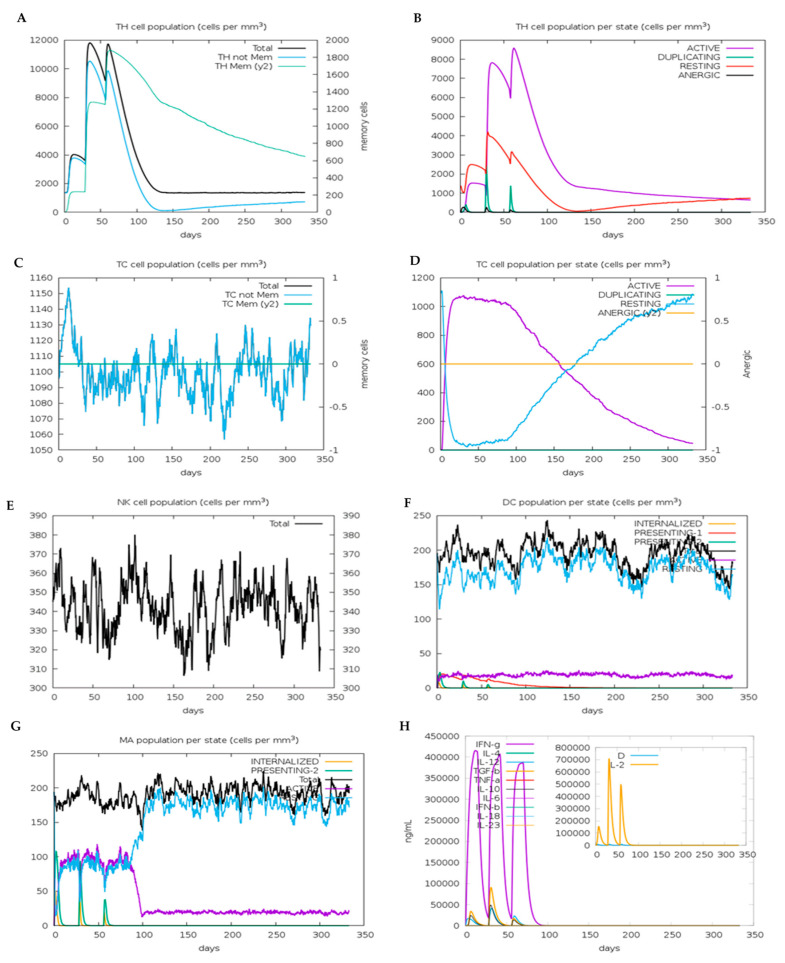
The immune simulation analysis of the VC: (**A**) Population of HTL. (**B**) The population of different states of HTL. (**C**) population of CTL. (**D**) Population of different states of CTL. (**E**) Population of NK cells. (**F**) Population of different states of dendritic cells. (**G**) Population of different states of macrophages. (**H**) Level of cytokines and interleukins produced in responses to vaccine.

**Figure 8 vaccines-09-01459-f008:**
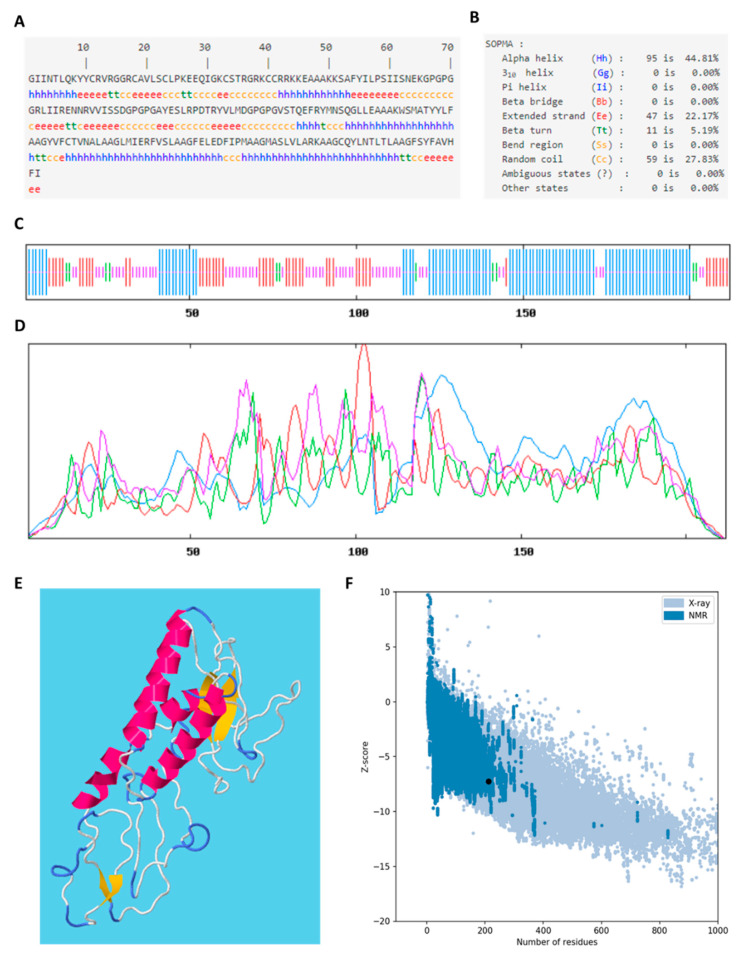
The secondary and tertiary structure of the VC: (**A**) Amino acid sequence and position of the secondary structure. (**B**) The global composition of the secondary structure of the VC. (**C**) Position of secondary structure within the protein sequence. The secondary structure is color-coded with blue—α-helix, red—extended strand, green—β-turn, and purple—random coil. (**D**) The propensity of each residue in adopting the secondary structure. (**E**) Tertiary structure as predicted by RAPTOR X (3D model) of the VC. (**F**) z-score value of the 3D model of the VC as calculated by ProSAweb is −7.25 (indicated by a black dot), which falls within the range of the z-score for the native proteins of similar size (212 aa).

**Figure 9 vaccines-09-01459-f009:**
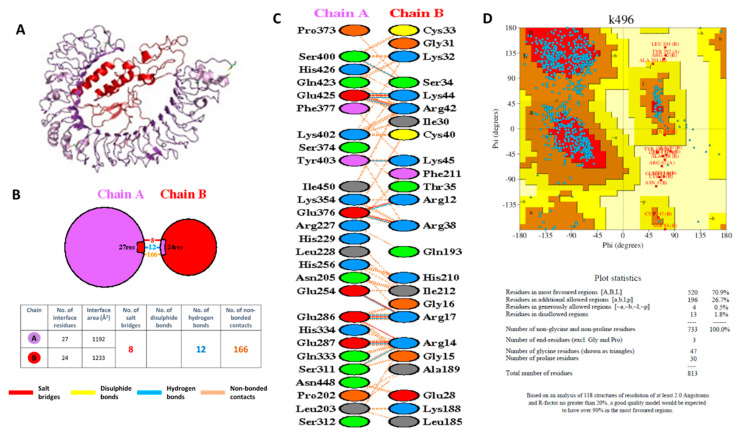
Interaction analysis of the VC and TLR4. (**A**) The tertiary structure model of the complex between TLR4 (purple) and the VC (red). (**B**) Diagram of interaction between the VC and TLR4 (red = salt bridges, blue = H-bonds, striped line = non-bonded contacts). (**C**) Residues involved in forming the complex. (**D**) Ramachandran plot of the interaction model showing the number of residues in the most favored region and less favored region.

**Figure 10 vaccines-09-01459-f010:**
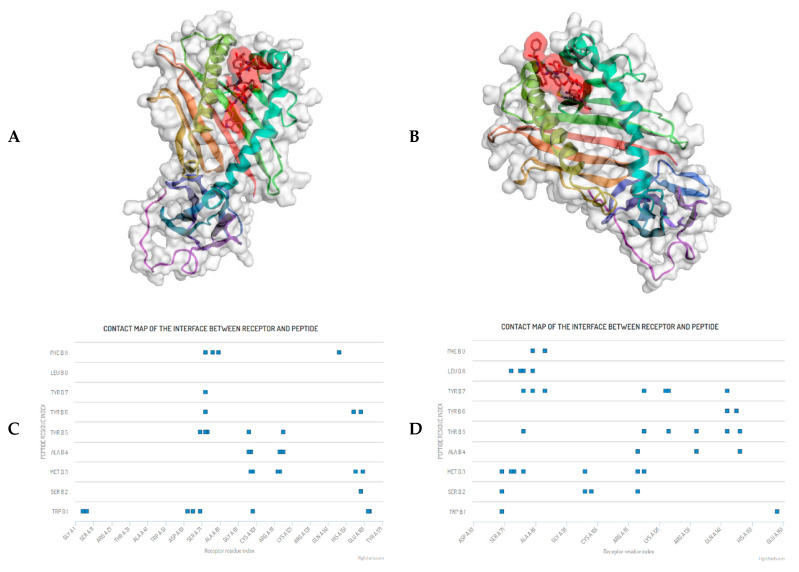
The best CABS-dock Modeling with default settings for peptide WSMATYYLF (stick structure in red color) with HLA-A*24:07 (**A**) and HLA-A*24:02 (**B**). The contact map between peptide and receptor for HLA-A*24:02 (**C**) and HLA-A*24:07 (**D**). CABS-dock server returns 10 top scored models of the protein-peptide complex. The best model prediction had high accuracy with average RMSD of 0.94 Å and 1.64 Å for HLA-A*24:07 and HLA-A*24:02, respectively. Moreover, the best model also had the highest cluster density score. HLA-A*24:07–WSMATYYLF complex had a higher accuracy compare to the HLA-A*24:02–WSMATYYLF complex.

**Table 1 vaccines-09-01459-t001:** The number of ORF1ab from SARS-CoV-2 variants included in the study. The sequences were retrieved from NCBI Virus SARS-CoV-2 Data Hub on 22 September 2021 in FASTA format. All sequences were of complete length between 7093 and 7096 amino acids and contained no ambiguous amino acid characters. The delta sequences were selected only from isolates collected between 31 July and 22 September 2021.

SARS-CoV-2 Variants	Number of Isolates
Alpha (B.1.1.7)	158
Beta (B.1.351)	374
Delta (B.1.617.2)	1157
Eta (B.1.525)	436
Gamma (P.1)	9
Iota (B.1.526)	24
Kappa (B.1.617.1)	148
Lambda (C.7)	286
Mu (B.1.621)	18

**Table 2 vaccines-09-01459-t002:** The number of SARS-CoV-2 immunogenic epitopes (T-cell assay positive) as reported in IEDB. The percentage of immunogenic epitopes over the possible number of 9-mer peptides generated by a protein was calculated. The number in bold indicates the percentage of the immunogenic epitopes over the total number of epitopes. IEDB is accessed on 16 September 2021.

Protein	Size (aa)	Number of Immunogenic Epitopes		
		Reported in IEDB (T-Cell Assay Positive)	% Immunogenic Epitopes Per Protein	% Immunogenic Epitopes Per Total Reported in IEDB
ORF1ab	7096	678	9.6	38.4
Spike	1273	578	4.5	32.7
ORF3a	275	88	32.0	5
Envelope	75	13	17.3	0.7
Membrane	222	131	59.0	7.4
ORF6	61	18	29.5	1.0
ORF7a	121	28	23.1	1.6
ORF7b	43	3	7.0	0.2
ORF8	121	37	30.6	2.1
Nucleocapsid	419	185	44.2	0.5
ORF10	38	8	21.0	0.5
Total epitopes		1767		

**Table 3 vaccines-09-01459-t003:** The number of experimentally known T-cell epitopes associated with HLA alleles of Indonesia (INA), Thailand (THA), and Germany (GER). The HLA alleles listed are predominant in the Indonesian and Thai populations (allele frequency ≥ 5%). The HLA alleles of the German population are the ones included in a study of T-cell responses to SARS-CoV-2 [25]. The number of immunogenic T-cell epitopes associated with the alleles were extracted from IEDB (accessed on 16 September 2021).

HLA alleles	Populations	ORF1ab T-Cell Epitopes			SARS-CoV-2 T-Cell Epitopes			% ORF1ab/SARS-CoV-2 Epitopes in T-Cell Assay
		Total	T-Cell Assay	HLA Assay	Total	T-Cell Assay	HLA Assay	
A*01:01	GER	54	48	12	96	85	21	56.47
A*02:01	GER INA	138	82	86	224	156	126	52.56
A*02:03	INA THA	0	0	0	0	0	0	
A*02:07	THA	0	0	0	0	0	0	
A*03:01	GER	42	17	33	69	37	45	45.95
A*11:01	GER INA THA	49	19	39	69	33	48	57.58
A*24:02	GER INA THA	64	45	29	129	100	47	45.00
A*24:07	INA THA	0	0	0	0	0	0	
A*33:03	INA THA	0	0	0	0	0	0	
A*34:01	INA	0	0	0	0	0	0	
B*07:02	GER	38	34	4	81	72	13	47.22
B*08:01	GER	26	25	1	56	52	4	48.08
B*13:01	THA	0	0	0	1	1	0	0.00
B*15:01	GER	34	29	5	56	44	12	65.91
B*15:02	INA THA	0	0	0	0	0	0	
B*15:13	INA	0	0	0	0	0	0	
B*15:21	INA	0	0	0	0	0	0	
B*18:01	INA THA	1	0	1	3	0	3	
B*35:05	INA	n.a.	n.a.	n.a.	n.a.	n.a.	n.a.	
B*38:02	INA	0	0	0	0	0	0	
B*40:01	GER THA	41	18	28	67	33	41	54.55
B*44:03	INA THA	11	11	0	25	25	0	44.00
B*46:01	THA	0	0	0	0	0	0	
B*58:01	INA THA	6	0	6	14	0	14	
DRB1*01:01	GER	8	4	7	61	8	59	50.00
DRB1*03:01	GER THA	1	0	1	28	20	15	0.00
DRB1*04:01	GER	24	2	24	156	5	156	40.00
DRB1*04:05	THA	1	0	1	39	0	39	
DRB1*07:01	GER INA THA	9	9	1	63	34	40	26.47
DRB1*09:01	THA	1	0	1	38	0	38	
DRB1*11:01	GER INA	1	0	1	42	12	32	0.00
DRB1*12:02	INA THA	0	0	0	3	3	0	0.00
DRB1*14:54	THA	0	0	0	0	0	0	
DRB1*15:01	GER INA THA	17	14	7	83	50	58	28.00
DRB1*15:02	INA THA	2	2	0	10	10	0	20.00
DRB1*16:02	INA THA	0	0	0	8	8	0	0.00

**Table 4 vaccines-09-01459-t004:** List of 9-mer peptides from ORF1ab predicted as promiscuous CTL epitopes.

Start Residue	Peptide	HLA Class I Alleles	Immunogenicity Score
295	FMGRIRSVY	HLA-A*01:01, HLA-A*29:01, HLA-B*15:01, HLA-B*15:02, HLA-B*15:12, HLA-B*15:13, HLA-B*15:21, HLA-B*15:25, HLA-B*15:32, HLA-B*35:01, HLA-B*35:05, HLA-B*35:30, HLA-B*46:01	0.1259
541	RVVRSIFSR	HLA-A*03:01, HLA-A*11:01, HLA-A*11:04, HLA-A*33:03, HLA-A*74:01	0.0318
611	WLTNIFGTV	HLA-A*02:01, HLA-A*02:03	0.2972
806	MVTNNTFTL	HLA-A*02:06, HLA-A*34:01, HLA-B*35:02, HLA-B*35:30, HLA-B*56:01, HLA-B*56:02, HLA-B*46:01	0.1578
899	WSMATYYLF ^b^	HLA-A*01:01, HLA-A*24:02, HLA-A*24:07, HLA-A*24:10, HLA-A*29:01, HLA-A*32:01, HLA-B*13:01, HLA-B*15:02, HLA-B*15:12, HLA-B*15:13, HLA-B*15:17, HLA-B*15:21, HLA-B*15:25, HLA-B*15:32, HLA-B*18:01, HLA-B*18:02, HLA-B*35:01, HLA-B*35:05, HLA-B*35:30, HLA-B*52:01, HLA-B*56:07, HLA-B*57:01, HLA-B*58:01, HLA-B*46:01	0.0071
1055	VVVNAANVY ^a^	HLA-A*26:01, HLA-B*15:01, HLA-B*15:02, HLA-B*15:12, HLA-B*15:21, HLA-B*15:25, HLA-B*15:32, HLA-B*35:01, HLA-B*46:01	0.1005
1140	HEVLLAPLL ^c^	HLA-B*13:01, HLA-B*18:01, HLA-B*18:02, HLA-B*37:01, HLA-B*38:02, HLA-B*40:01, HLA-B*40:02, HLA-B*40:06, HLA-B*41:01, HLA-B*44:03	0.0124
1247	FLTENLLLY ^b^	HLA-A*01:01, HLA-A*26:01, HLA-A*29:01	0.0808
1254	LYIDINGNL	HLA-A*24:02, HLA-A*24:07, HLA-A*24:10	0.2138
1269	LVSDIDITF ^a^	HLA-B*15:02, HLA-B*15:13, HLA-B*15:17, HLA-B*15:21, HLA-B*35:01, HLA-B*35:02, HLA-B*35:05, HLA-B*35:30, HLA-B*57:01, HLA-B*58:01, HLA-B*46:01	0.2541
1366	ILGTVSWNL ^b^	HLA-A*02:01, HLA-A*02:07	0.1177
1674	YLATALLTL ^a,b^	HLA-A*02:01, HLA-A*02:03, HLA-A*02:06, HLA-A*02:07, HLA-B*46:01	0.0927
2175	LLQLCTFTR	HLA-A*33:03, HLA-A*74:01	0.0568
2327	FLAYILFTR	HLA-A*33:03, HLA-A*74:01	0.2496
2331	ILFTRFFYV ^a,b^	HLA-A*02:01, HLA-A*02:03, HLA-A*02:06, HLA-A*74:01, HLA-B*08:01, HLA-A*02:07	0.3343
2350	FSYFAVHFI	HLA-B*51:01, HLA-B*51:02, HLA-B*52:01	0.2893
2597	FSSTFNVPM	HLA-B*15:10, HLA-B*15:21, HLA-B*35:01, HLA-B*35:05, HLA-B*35:30, HLA-B*56:02, HLA-B*46:01	0.1216
2629	LSTFISAAR	HLA-A*33:03, HLA-A*34:01, HLA-A*74:01	0.1602
2784	AIFYLITPV ^b,c^	HLA-A*02:01, HLA-A*02:03, HLA-A*02:06, HLA-A*34:01, HLA-A*02:07	0.1750
2786	FYLITPVHV ^a^	HLA-A*24:02, HLA-A*24:07, HLA-A*24:10	0.2114
2787	YLITPVHVM ^a^	HLA-A*02:01, HLA-A*02:03, HLA-A*02:06, HLA-A*26:01, HLA-B*15:01, HLA-B*15:02, HLA-B*15:10, HLA-B*15:12, HLA-B*15:21, HLA-B*15:25, HLA-B*15:32, HLA-B*35:01, HLA-A*02:07, HLA-B*46:01	0.1617
2883	FLPRVFSAV ^a,b^	HLA-A*02:01, HLA-A*02:03, HLA-A*02:06, HLA-B*08:01, HLA-A*02:07	0.0821
3059	LAYYFMRFR ^a^	HLA-A*33:03, HLA-A*74:01	0.0559
3060	AYYFMRFRR	HLA-A*33:03, HLA-A*74:01	0.1234
3076	VVAFNTLLF	HLA-A*24:07, HLA-A*29:01	0.1449
3121	FLAHIQWMV ^a,b^	HLA-A*02:01, HLA-A*02:03, HLA-A*02:06, HLA-A*02:07	0.1502
3137	FWITIAYII ^d^	HLA-A*24:02, HLA-A*24:07, HLA-A*24:10	0.3233
3152	FYWFFSNYL	HLA-A*24:02, HLA-A*24:07, HLA-A*24:10	0.1404
3466	VLAWLYAAV ^a,b^	HLA-A*02:01, HLA-A*02:03, HLA-A*02:06, HLA-A*02:07	0.2772
3481	FLNRFTTTL ^a,b^	HLA-A*02:01, HLA-A*02:03, HLA-A*02:06, HLA-B*08:01, HLA-A*02:07, HLA-B*46:01	0.2560
3582	LLLTILTSL ^b,c^	HLA-A*02:01, HLA-A*02:03, HLA-A*02:06, HLA-B*08:01, HLA-A*02:07	0.0907
3605	LYENAFLPF	HLA-A*24:02, HLA-A*24:07, HLA-A*24:10	0.1584
3652	VYMPASWVM ^a,b^	HLA-A*24:02, HLA-A*24:07, HLA-A*24:10	0.0253
3684	YASAVVLLI ^a,c^	HLA-B*51:01, HLA-B*51:02, HLA-B*52:01, HLA-B*56:07, HLA-B*58:01	0.0489
3692	ILMTARTVY ^a^	HLA-A*29:01, HLA-B*15:01, HLA-B*15:02, HLA-B*15:12, HLA-B*15:21, HLA-B*15:25, HLA-B*15:32, HLA-B*35:05, HLA-B*35:30, HLA-B*46:01	0.1258
3752	FLARGIVFM ^a,b,c^	HLA-A*02:01, HLA-A*02:03, HLA-A*02:06, HLA-A*02:07	0.3263
4030	TMLFTMLRK ^b^	HLA-A*03:01, HLA-A*11:01, HLA-A*11:04, HLA-A*74:01	0.0076
4265	VLSFCAFAV ^b^	HLA-A*02:01, HLA-A*02:07	0.1701
4513	YTMADLVYA ^b^	HLA-A*02:01, HLA-A*02:06, HLA-A*02:07	0.0262
4656	YIKWDLLKY	HLA-A*01:01, HLA-A*26:01, HLA-A*29:01, HLA-B*15:02, HLA-B*15:12, HLA-B*15:21, HLA-B*46:01	0.0287
4698	ILHCANFNV ^a^	HLA-A*02:01, HLA-A*02:03, HLA-A*02:06, HLA-A*02:07	0.0833
4723	KIFVDGVPF	HLA-A*32:01, HLA-B*15:01, HLA-B*15:02, HLA-B*15:25, HLA-B*15:32	0.1614
4846	YYRYNLPTM	HLA-A*24:02, HLA-A*24:10	0.0097
4862	FVVEVVDKY ^a^	HLA-A*26:01, HLA-A*29:01, HLA-A*34:01, HLA-B*15:21, HLA-B*35:01, HLA-B*35:30, HLA-B*46:01	0.0859
5024	MASLVLARK ^a^	HLA-A*03:01, HLA-A*11:01, HLA-A*11:04, HLA-A*30:01, HLA-A*33:03, HLA-A*34:01, HLA-A*74:01	0.0282
5132	FVNEFYAYL ^a^	HLA-A*02:01, HLA-A*02:03, HLA-A*02:06, HLA-A*26:01, HLA-A*34:01, HLA-A*02:07, HLA-B*46:01	0.2400
5245	LMIERFVSL ^a^	HLA-A*02:01, HLA-A*02:03, HLA-A*02:06, HLA-A*32:01, HLA-B*08:01, HLA-B*15:01, HLA-B*15:02, HLA-B*15:10, HLA-B*15:12, HLA-B*15:21, HLA-B*15:25, HLA-B*15:32, HLA-B*35:02, HLA-B*37:01, HLA-B*38:02, HLA-B*48:01, HLA-A*02:07, HLA-B*46:01	0.2427
5247	IERFVSLAI	HLA-B*13:01, HLA-B*37:01, HLA-B*40:01, HLA-B*40:02, HLA-B*40:06, HLA-B*41:01, HLA-B*44:03, HLA-B*52:01	0.0326
5250	FVSLAIDAY	HLA-A*01:01, HLA-A*26:01, HLA-A*29:01, HLA-A*34:01, HLA-B*15:02, HLA-B*15:21, HLA-B*35:01, HLA-B*35:05, HLA-B*35:30, HLA-B*46:01	0.1401
5273	HLYLQYIRK ^b^	HLA-A*03:01, HLA-A*11:01, HLA-A*11:04, HLA-A*74:01	0.0139
5614	FAIGLALYY ^a,c^	HLA-A*01:01, HLA-A*26:01, HLA-A*29:01, HLA-B*15:13, HLA-B*15:21, HLA-B*35:01, HLA-B*35:05, HLA-B*35:30, HLA-B*58:01, HLA-B*46:01	0.0918
5678	YVFCTVNAL ^a^	HLA-A*02:01, HLA-A*02:03, HLA-A*02:06, HLA-A*26:01, HLA-A*34:01, HLA-B*07:02, HLA-B*07:05, HLA-B*15:02, HLA-B*15:10, HLA-B*15:21, HLA-B*35:01, HLA-B*35:02, HLA-B*35:05, HLA-B*35:30, HLA-B*38:02, HLA-B*48:01, HLA-B*56:01, HLA-B*56:02, HLA-A*02:07, HLA-B*46:01	0.0778
6070	FKHLIPLMY	HLA-A*29:01, HLA-B*18:02	0.0065
6108	VLWAHGFEL ^a^	HLA-A*02:01, HLA-A*02:06, HLA-A*02:07	0.3320
6506	FELWAKRNI	HLA-B*40:01, HLA-B*40:02, HLA-B*40:06, HLA-B*41:01	0.0943
6585	FRNARNGVL	HLA-B*15:10, HLA-B*27:06	0.1343
6700	HLLIGLAKR	HLA-A*33:03, HLA-A*74:01	0.0599
6714	FELEDFIPM ^b^	HLA-B*13:01, HLA-B*15:10, HLA-B*18:01, HLA-B*18:02, HLA-B*37:01, HLA-B*38:02, HLA-B*40:01, HLA-B*40:02, HLA-B*40:06, HLA-B*41:01, HLA-B*44:03, HLA-B*48:01	0.3348
6748	LLLDDFVEI ^a,b,c^	HLA-A*02:01, HLA-A*02:03, HLA-A*02:06, HLA-B*52:01, HLA-A*02:07	0.2439
6848	CQYLNTLTL	HLA-B*13:01, HLA-B*15:10, HLA-B*27:06, HLA-B*37:01, HLA-B*38:02, HLA-B*48:01, HLA-B*52:01	0.0312
6850	YLNTLTLAV ^a,b^	HLA-A*02:01, HLA-A*02:03, HLA-A*02:06, HLA-A*02:07	0.0762
6885	WLPTGTLLV	HLA-A*02:01, HLA-A*02:03, HLA-A*02:06, HLA-A*02:07	0.0892
6978	YKLMGHFAW	HLA-B*18:01, HLA-B*18:02	0.0048
7019	YVMHANYIF ^a^	HLA-A*24:02, HLA-A*24:07, HLA-B*15:02, HLA-B*15:13, HLA-B*15:21, HLA-B*35:01, HLA-B*35:05, HLA-B*35:30, HLA-B*56:02, HLA-B*46:01	0.0822
7026	IFWRNTNPI	HLA-A*24:02, HLA-A*24:07, HLA-A*24:10	0.1423

^a^ The peptide has been experimentally proven by T-cell assay and reported in IEDB; ^b^ The peptide has been experimentally proven by HLA binding and reported in IEDB; ^c^ Peptide has some degree of homology with human self-peptide; ^d^ The peptide existed only in 9.61% of delta variant isolates, whereas the rest had the mutant peptide ^3137^SWITIAYII^3145.^

**Table 5 vaccines-09-01459-t005:** List of 15-mer peptides predicted as promiscuous HTL epitopes and their IFNγ score.

Start Residue	Epitope Sequence	HLA DRB1 Alleles	IFNγ Score
447	NDNLLEILQKEKVNI	DRB1*12:02, DRB1*14:01, DRB1*14:04, DRB1*14:05, DRB1*14:54,	0.1311
448	DNLLEILQKEKVNIN	DRB1*12:02, DRB1*14:01, DRB1*14:04, DRB1*14:05, DRB1*14:54,	0.0556
554	TAQNSVRVLQKAAIT	DRB1*12:02, DRB1*14:01, DRB1*14:04, DRB1*14:05, DRB1*14:54,	0.0684
736	PKEIIFLEGETLPTE	DRB1*01:01, DRB1*12:02, DRB1*15:01, DRB1*15:02, DRB1*16:02,	0.0771
1054	PTVVVNAANVYLKHG	DRB1*13:02, DRB1*14:01, DRB1*14:04, DRB1*14:54, DRB1*15:01, DRB1*15:02, DRB1*16:02	0.0917
1187	VSSFLEMKSEKQVEQ	DRB1*04:01, DRB1*04:03, DRB1*04:05, DRB1*04:06, DRB1*10:01,	0.0899
1211	VKPFITESKPSVEQR	DRB1*08:03, DRB1*11:01, DRB1*13:02, DRB1*14:05, DRB1*14:07,	0.3157
1349	CKSAFYILPSIISNE	DRB1*01:01, DRB1*04:01, DRB1*04:05, DRB1*08:03, DRB1*10:01, DRB1*11:01, DRB1*15:02, DRB1*16:02,	0.2898
1350	KSAFYILPSIISNEK ^a^	DRB1*01:01, DRB1*04:01, DRB1*04:03, DRB1*04:05, DRB1*04:06, DRB1*08:03, DRB1*10:01, DRB1*11:01, DRB1*12:02, DRB1*15:02, DRB1*16:02,	0.3378
1355	ILPSIISNEKQEILG	DRB1*13:02, DRB1*14:01, DRB1*14:04, DRB1*14:05, DRB1*14:07, DRB1*14:54,	0.4244
1356	LPSIISNEKQEILGT	DRB1*13:02, DRB1*14:01, DRB1*14:04, DRB1*14:05, DRB1*14:07, DRB1*14:54,	0.3025
1357	PSIISNEKQEILGTV	DRB1*13:02, DRB1*14:01, DRB1*14:04, DRB1*14:05, DRB1*14:54,	0.5074
2944	AYESLRPDTRYVLMD	DRB1*03:01, DRB1*14:01, DRB1*14:04, DRB1*14:05, DRB1*14:54,	0.3078
2945	YESLRPDTRYVLMDG	DRB1*03:01, DRB1*13:02, DRB1*14:01, DRB1*14:04, DRB1*14:05, DRB1*14:07, DRB1*14:54,	0.1649
2958	DGSIIQFPNTYLEGS	DRB1*04:02, DRB1*13:02, DRB1*15:01, DRB1*15:02, DRB1*16:02,	0.2103
3815	VSTQEFRYMNSQGLL	DRB1*01:01, DRB1*07:01, DRB1*09:01, DRB1*15:02, DRB1*16:02,	0.0976
3944	IASEFSSLPSYAAFA	DRB1*01:01, DRB1*04:01, DRB1*10:01, DRB1*15:02, DRB1*16:02,	0.0754
3945	ASEFSSLPSYAAFAT	DRB1*01:01, DRB1*04:01, DRB1*10:01, DRB1*15:02, DRB1*16:02,	0.3973
3951	LPSYAAFATAQEAYE	DRB1*04:01, DRB1*04:03, DRB1*04:05, DRB1*04:06, DRB1*08:03,	0.0518
4457	LIDSYFVVKRHTFSN	DRB1*08:03, DRB1*11:01, DRB1*13:02, DRB1*14:01, DRB1*14:04, DRB1*14:07, DRB1*14:54,	0.1304
4458	IDSYFVVKRHTFSNY	DRB1*08:03, DRB1*11:01, DRB1*13:02, DRB1*14:01, DRB1*14:04, DRB1*14:07, DRB1*14:54,	0.1870
4560	NPDILRVYANLGERV	DRB1*04:02, DRB1*08:03, DRB1*12:02, DRB1*15:01, DRB1*15:02, DRB1*16:02,	0.2299
4561	PDILRVYANLGERVR ^a^	DRB1*04:02, DRB1*08:03, DRB1*13:02, DRB1*15:01, DRB1*15:02, DRB1*16:02,	0.2616
4761	KELLVYAADPAMHAA	DRB1*04:01, DRB1*04:02, DRB1*15:01, DRB1*15:02, DRB1*16:02,	0.2258
4830	KHFFFAQDGNAAISD	DRB1*01:01, DRB1*04:01, DRB1*10:01, DRB1*14:07, DRB1*16:02,	0.4401
4933	QMNLKYAISAKNRAR	DRB1*08:03, DRB1*10:01, DRB1*11:01, DRB1*13:02, DRB1*14:05, DRB1*14:07,	0.4044
4934	MNLKYAISAKNRART	DRB1*08:03, DRB1*10:01, DRB1*11:01, DRB1*13:02, DRB1*14:05, DRB1*14:07,	0.4019
4935	NLKYAISAKNRARTV	DRB1*08:03, DRB1*11:01, DRB1*13:02, DRB1*14:05, DRB1*14:07,	0.5938
5019	PNMLRIMASLVLARK ^a^	DRB1*01:01, DRB1*12:02, DRB1*14:04, DRB1*15:01, DRB1*15:02, DRB1*16:02,	0.3914
5717	AKHYVYIGDPAQLPA	DRB1*04:01, DRB1*04:03, DRB1*04:05, DRB1*04:06, DRB1*08:03, DRB1*10:01, DRB1*16:02,	0.1673
5775	TVSALVYDNKLKAHK	DRB1*03:01, DRB1*11:01, DRB1*13:02, DRB1*14:01, DRB1*14:04, DRB1*14:05, DRB1*14:07, DRB1*14:54,	0.3517
5776	VSALVYDNKLKAHKD ^a^	DRB1*03:01, DRB1*08:03, DRB1*11:01, DRB1*13:02, DRB1*14:01, DRB1*14:04, DRB1*14:05, DRB1*14:07, DRB1*14:54,	0.2560
5777	SALVYDNKLKAHKDK	DRB1*03:01, DRB1*13:02, DRB1*14:01, DRB1*14:04, DRB1*14:05, DRB1*14:07, DRB1*14:54,	0.4910
5834	VFISPYNSQNAVASK	DRB1*01:01, DRB1*04:01, DRB1*04:02, DRB1*10:01, DRB1*15:01, DRB1*15:02,	0.2236
6046	PTGYVDTPNNTDFSR	DRB1*04:01, DRB1*04:03, DRB1*04:05, DRB1*04:06, DRB1*08:03,	0.0690
6454	LENVAFNVVNKGHFD	DRB1*13:02, DRB1*14:01, DRB1*14:04, DRB1*14:05, DRB1*14:07, DRB1*14:54,	0.0787
6726	TVKNYFITDAQTGSS	DRB1*01:01, DRB1*04:01, DRB1*07:01, DRB1*09:01, DRB1*10:01, DRB1*16:02,	0.0871
7075	KGRLIIRENNRVVIS	DRB1*04:02, DRB1*13:02, DRB1*14:01, DRB1*14:04, DRB1*14:05, DRB1*14:54, DRB1*15:01, DRB1*15:02	0.7895
7076	GRLIIRENNRVVISS	DRB1*04:02, DRB1*13:02, DRB1*14:01, DRB1*14:04, DRB1*14:05, DRB1*14:54, DRB1*15:01, DRB1*15:02	0.7985
7077	RLIIRENNRVVISSD	DRB1*13:02, DRB1*14:01, DRB1*14:05, DRB1*14:07, DRB1*14:54,	0.8026

^a^ The peptide has been experimentally proven by T-cell assay and reported in IEDB.

**Table 6 vaccines-09-01459-t006:** Comparison of binding affinity between the ancestral (FWITIAYII) and mutant (SWITIAYII) peptide. The calculation was made using NetCTLpan 1.1., which also predicted the peptide processing (proteasomal cleavage and TAP transport efficiency) inside the cell.

Peptide	Allele	HLA	TAP	Cle	Comb	Aff(nM)	%Rank
FWITIAYII	HLA-A*24:02	0.604	0.566	0.463	0.722	250.54	0.8
SWITIAYII	HLA-A*24:02	0.65	0.884	0.617	0.811	35.49	0.3
FWITIAYII	HLA-A*24:07	0.497	0.566	0.463	0.615	220.88	0.8
SWITIAYII	HLA-A*24:07	0.591	0.884	0.617	0.752	40.82	0.15
FWITIAYII	HLA-A*24:10	0.8	0.566	0.463	0.918	61.69	0.8
SWITIAYII	HLA-A*24:10	0.848	0.884	0.617	1.009	14.15	0.4

**Table 7 vaccines-09-01459-t007:** SARS-CoV-2 peptides with sequence homologous with human peptides (underlined). The HLA alleles presenting the human peptides as revealed by NetCTLpan analysis are indicated. Some of these peptides were curated in IEDB and had been confirmed by T-cell assay, HLA assay, or both.

Start	SARS-CoV-2 Peptide	Human Peptides	Human Proteins	HLA Allele Presenting the Human Peptide	IEDB Confirmation Assay
1140	HEVLLAPLL	AEVLLAPLL	HSVI binding protein (AAF76892.1)	HLA-B*37:01, HLA-B*38:02, HLA-B*40:01, HLA-B*40:02, HLA-B*40:06, HLA-B*41:01, HLA-B*44:03, HLA-B*13:01	n.a.
2784	AIFYLITPV	AIFYLITLV	olfactory receptor, family 2, subfamily W, member 1, isoform CRA_b (EAX03180.1)	HLA-A*02:01, HLA-A*02:03, HLA-A*02:06	T-cell assay and HLA assay
3582	LLLTILTSL	LLLTILTRP	hCG2023968 (EAW49626.1)	non binder	HLA assay
3684	YASAVVLLI	VASAVVLLG	molybdenum cofactor biosynthesis protein 1 isoform 7 (NP_001345459.1)	non-binder	T-cell assay
3752	FLARGIVFM	XCARGIVFM	immunoglobulin heavy chain junction region (MOL38621.1)	cannot generate a similar peptide, since the sequence is at the N-terminal end of the protein.	T-cell assay and HLA assay
5614	FAIGLALYY	SYIGLALYY	immunoglobulin heavy chain junction region (MOJ91547.1)	HLA-A*29:01	T cell assay
6748	LLLDDFVEI	IALDDFVEI	Wolfram syndrome 1 (wolframin), isoform CRA_a (EAW82396.1)	HLA-A*02:06, HLA-B*52:01	T-cell assay and HLA assay

**Table 8 vaccines-09-01459-t008:** Seven CTL and five HTL epitopes chosen for the VC and the population coverage. Epitopes fulfilled the criteria such as highest percentile rank, high promiscuity, high immunogenicity, high IFNγ induction ability, conservancy across all variants, low entropy value, and the absence of homology with human peptides.

Start Residue	Peptide and Entropy *	HLA Alleles Bind to the Peptides	Population Coverage			
			Indonesia	Thailand	Germany	World
899	WSMATYYLF (0.083)	HLA-A*01:01, HLA-A*24:02, HLA-A*24:07, HLA-A*24:10, HLA-A*29:01, HLA-A*32:01, HLA-B*13:02, HLA-B*15:02, HLA-B*15:12, HLA-B*15:13, HLA-B*15:17, HLA-B*15:21, HLA-B*15:25, HLA-B*15:32, HLA-B*18:01, HLA-B*18:02, HLA-B*35:01, HLA-B*35:05, HLA-B*35:30, HLA-B*52:01, HLA-B*56:07, HLA-B*57:01, HLA-B*58:01, HLA-B*46:01	94.80	77.44;	66.25;	64.13
5678	YVFCTVNAL (0.026)	HLA-A*02:01, HLA-A*02:03, HLA-A*02:06, HLA-A*26:01, HLA-A*34:01, HLA-B*07:02, HLA-B*07:05, HLA-B*15:02, HLA-B*15:10, HLA-B*15:21, HLA-B*35:01, HLA-B*35:02, HLA-B*35:05, HLA-B*35:30, HLA-B*38:02, HLA-B*48:01, HLA-B*56:01, HLA-B*56:02, HLA-A*02:07, HLA-B*46:01	77.39	75.05	72.07	65.66
5245	LMIERFVSL (0.000)	HLA-A*02:01, HLA-A*02:03, HLA-A*02:06, HLA-A*32:01, HLA-B*08:01, HLA-B*15:01, HLA-B*15:02, HLA-B*15:10, HLA-B*15:12, HLA-B*15:21, HLA-B*15:25, HLA-B*15:32, HLA-B*35:02, HLA-B*37:01, HLA-B*38:02, HLA-B*48:01, HLA-A*02:07, HLA-B*46:01	63.65	74.26	71.42	63.19
6714	FELEDFIPM (0.037)	HLA-B*13:01, HLA-B*13:02, HLA-B*15:10, HLA-B*18:01, HLA-B*18:02, HLA-B*37:01, HLA-B*38:02, HLA-B*40:01, HLA-B*40:02, HLA-B*40:06, HLA-B*41:01, HLA-B*44:03, HLA-B*48:01	51.64	46.46	35.87	35.59
5024	MASLVLARK (0.000)	HLA-A*03:01, HLA-A*11:01, HLA-A*11:04, HLA-A*30:01, HLA-A*33:03, HLA-A*34:01, HLA-A*74:01	67.42	55.82	40.12	40.42
6848	CQYLNTLTL (0.000)	HLA-B*13:02, HLA-B*15:10, HLA-B*27:06, HLA-B*37:01, HLA-B*38:02, HLA-B*48:01, HLA-B*52:01	21.20	20.90	10.15	13.16
2350	FSYFAVHFI (0.027)	HLA-B*51:01, HLA-B*52:01	8.29	13.51	12.13	10.26
1350	KSAFYILPSIISNEK (0.0283; 0.027; 0.023; 0.015; 0.015; 0.013; 0.013)	DRB1*01:01, DRB1*04:01, DRB1*04:03, DRB1*04:05, DRB1*04:06, DRB1*08:03, DRB1*10:01, DRB1*11:01, DRB1*12:02, DRB1*15:02, DRB1*16:02	91.13	74.99	47.87	47.60
7076	GRLIIRENNRVVISS (0.000; 0.000; 0.000; 0.000; 0.000; 0.000; 0.000)	DRB1*04:02, DRB1*13:02, DRB1*14:01, DRB1*14:04, DRB1*14:05, DRB1*14:54, DRB1*15:01, DRB1*15:02	53.28	50.97	40.57	37.72
7077	RLIIRENNRVVISSD (0.000; 0.000; 0.000; 0.000; 0.000; 0.000; 0.000)	DRB1*13:02, DRB1*14:01, DRB1*14:05, DRB1*14:07, DRB1*14:54	4.18	11.27	13.43	16.78
2944	AYESLRPDTRYVLMD (0.068; 0.045; 0.039; 0.0505; 0.027; 0.028; 0.026)	DRB1*03:01, DRB1*14:01, DRB1*14:04, DRB1*14:05, DRB1*14:54	10.59	23.69	25.46	27.58
3815	VSTQEFRYMNSQGLL (0.000; 0.000; 0.000; 0.000; 0.000; 0.000; 0.129)	DRB1*01:01, DRB1*07:01, DRB1*09:01, DRB1*15:02, DRB1*16:02	63.57	63.45	41.63	38.08
Epitope set			100.00	100.00	99.98	99.88

* The entropy value in the bracket. For HTL epitopes the entropy values are for seven possible 9-mer core-peptides.

**Table 9 vaccines-09-01459-t009:** Population coverage of the 12 chosen epitopes. A projected population coverage, average number of epitope hits/HLA combinations recognized by the population, and minimum number of epitope hits/HLA combinations recognized by 90% of the population.

Population/Area	Class I			Class II			Class Combined		
	Coverage ^a^	Average_Hit ^b^	pc90 ^c^	Coverage ^a^	Average_Hit ^b^	pc90 ^c^	Coverage ^a^	Average_Hit ^b^	pc90 ^c^
Germany	99.75%	3.89	2.54	93.25%	1.81	1.09	99.98%	5.7	4.03
Indonesia	100.0%	5.66	4.1	99.68%	2.68	1.46	100.0%	8.35	6.19
Thailand	99.76%	4.82	2.9	98.69%	2.63	1.48	100.0%	7.45	5.08
World	98.77%	3.65	2.08	90.66%	1.82	1.02	99.88%	5.47	3.38
Average	99.57	4.5	2.9	95.57	2.23	1.26	99.97	6.74	4.67
Standard deviation	0.47	0.8	0.75	3.75	0.42	0.21	0.05	1.2	1.07

^a^ Projected population coverage; ^b^ average number of epitope hits/HLA combinations recognized by the population; ^c^ minimum number of epitope hits/HLA combinations recognized by 90% of the population.

## Data Availability

Not applicable.

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
