# Peer review of "Immunoinformatics Analysis of SARS-CoV-2 ORF1ab Polyproteins to Identify Promiscuous and Highly Conserved T-Cell Epitopes to Formulate Vaccine for Indonesia and the World Population"

_vaccines, 2021, doi:10.3390/vaccines9121459_

Round 1
Reviewer 1 Report
This MS describes an in silico analysis that focuses on the identification of promiscuous and conserved T-cell epitopes of SARS-CoV2 ORF1ab. By using multiple immunoinformatic and structural modelling tools, this study identified a promiscuous peptide 899WSMATYYLF907 from SARS-CoV2 ORF1ab that may be presented as a predominant human leukocyte antigen in Indonesia (i.e. HLA-A*24:07). The results of this study have potential application to the development of peptide-based vaccine against SARS-CoV2. Specific comments are as follows:
- The top of Figure 1 seems to be cut off. The quality of Figure 8A, B and E is not good, please improve.
- (Abstract) Please provide full names of CTL, HTL, HLA. Also, please provide the abbreviation “VC” after “vaccine construct”.
- (Lines 359 and 839) Please describe how the Z-score value was evaluated. In what Z-score range is the model quality acceptable?
- (Lines 27-29) “The most promiscuous peptide 899WSMATYYLF907 was confirmed to be presented by HLA-A*24:07, the most predominant allele in Indonesia, via docking simulation”. Are there any experimental evidences for the binding of HLA-A*24:07 to this peptide? If not, please adjust the statement of this sentence.
- The major concern of this study is that all results are only based on an in silico analysis. It would be desirable to strengthen their investigation by adding experimental data such as HLA binding assays.
- In my opinion, SARS-CoV2 ORF1ab may not be a good target for vaccine development. The products of this ORF are non-structural proteins of SARS-CoV2. Viral non-structural proteins may only appear inside the infected cells for a very short time. Generally, although viral non-structural protein may play an important role in virus replication, it is difficult to become an immune antigen. Is there any successful peptide based vaccine (or other type of vaccine) on the market that targets viral non-structural proteins?
Author Response
Responses to Reviewer 1 comments
We thank Reviewer 1 for the valuable input and we would like to address some of the comments and questions below.
Point 1
Reviewer 1: The top of Figure 1 seems to be cut off. The quality of Figure 8A, B and E is not good, please improve.
M Gustiananda: Thank you for the suggestions. Kindly find that we have replaced Figure 1 so that the top part is not cut off and that the size of Figure 8A, 8B, and 8E has been adjusted.
Point 2
R1: (Abstract) Please provide full names of CTL, HTL, HLA. Also, please provide the abbreviation “VC” after “vaccine construct”.
MG: Thank you, for the suggestion. full names of the CTL, HTL, and HLA have been provided at the first instance that the term appear in the text. The abbreviation “VC” after “vaccine construct’ has also been provided. Kindly note that the abstract has been further revised to ensure that word count does not exceed 200 words.
Point 3
R1: (Lines 359 and 839) Please describe how the Z-score value was evaluated. In what Z-score range is the model quality acceptable?
MG: we thank the reviewer to point out about the z-score. The z-score value mentioned in Figure 8F and in the text (lines 359 and 839) was generated by the tool called ProSA web (Wiederstein & Sippl, 2007). The following text explaining how the z-score value was evaluated has been inserted in the new text
Lines 352-363
“….Tertiary structure generated by RAPTOR X was then validated by using ProSA-web (https://prosa.services.came.sbg.ac.at/prosa.php) [50]. The PDB structure generated by RAPTOR X was used as an input file for ProSA-web which will compare the predicted 3D model of the VC with the existing proteins structure in PDB database that were generated experimentally either by x-ray crystallography or NMR. ProSA-web then calculated the z-score which represents the quality index of the model. Graphically, the z-score value is displayed in a plot that contains the z-scores of all experimentally determined protein chains in PDB database, where dark blue and light blue area rep-resents NMR and x-ray structures, respectively. The z-score value of the predicted model is displayed as a black dot in the graph and the model quality is acceptable if it falls within the range of scores typically found for native proteins of similar size….”
Lines 841-843
“…..Since the size of the VC is 212 amino acids, the z-score should fall between -1 and -10, according to the plot. The z-score of VC is -7.25 which fell within the range of con-formational parameters of native proteins.….”
Lines 849-850
“……(F). z-score value of the 3D model of VC as calculated by ProSA -web is -7.25 (indicated by a black dot) which fell within the range of the z-score for the native proteins of similar size (212 aa)….”
Point 4
R1: (Lines 27-29) “The most promiscuous peptide 899WSMATYYLF907 was confirmed to be presented by HLA-A*24:07, the most predominant allele in Indonesia, via docking simulation”. Are there any experimental evidences for the binding of HLA-A*24:07 to this peptide? If not, please adjust the statement of this sentence.
MG: Thank you for addressing this issue. It is correct that there is no experimental evidence for the binding of HLA-A*24:07 with the 899WSMATYYLF907. The sentence has been adjusted into:
Lines 26-28
“The most promiscuous peptide 899WSMATYYLF907 was shown via docking simulation to interact well with HLA-A*24:07, the most predominant allele in Indonesia”.
Point 5
R1: The major concern of this study is that all results are only based on an in silico analysis. It would be desirable to strengthen their investigation by adding experimental data such as HLA binding assays.
MG: Thank you for addressing the major concern related to this study. We acknowledge the limitation of this study is that everything has been done in-silico and the wet lab experiment to verify the results of prediction was not yet carried out. The in-silico analysis is needed especially for HLA alleles that are less studied such as HLA-A*24:07 in order to get some information which peptides likely to bind to this HLA. HLA-A*24:07 (allele frequency 0.26) is the most predominant allele in Indonesia (population 270 million). There is an urgent need to generate the experimental data on the peptide binding to this HLA. While the experimental part is beyond the scope of this research paper, the data generated from this current study can help guide the choice of peptides to be tested further in the experiment, not only the HLA-peptide binding investigation but also T-cell assay involving ELISPOT screening using PBMC samples taken from COVID-19 convalescent individuals from Indonesia. By narrowing down the number of peptides to be tested in the experiment, the cost, time, and the amount of blood samples needed in the experiment can also be reduced significantly.
Point 6
R1: In my opinion, SARS-CoV2 ORF1ab may not be a good target for vaccine development. The products of this ORF are non-structural proteins of SARS-CoV2. Viral non-structural proteins may only appear inside the infected cells for a very short time. Generally, although viral non-structural protein may play an important role in virus replication, it is difficult to become an immune antigen. Is there any successful peptide based vaccine (or other type of vaccine) on the market that targets viral non-structural proteins?
MG: we appreciate the reviewer opinion about the suitability of the ORF1ab as vaccine target. It is correct that none of the COVID-19 vaccines available on the market today use ORF1ab as target antigen. This is due to the fact that the current vaccines mainly focus on the induction of humoral immunity, where antibody is the effector molecule. The antibody generated should target structural proteins on the surface of the virus to block the infection. In this current study we addressed the needs for vaccines that induce cell-mediated immunity, where T-cells are the effector component of the immune responses. T-cells will recognize infected cells based on the presentation of viral peptides by HLA molecules. In order for T-cells to see the antigen, the antigens do not need to appear in a long time inside the infected cells, this requirement is probably less important than the requirement for the proteins to be able to undergo MHC Class I pathway of the antigen processing inside the cells, such as proteasomal cleavage, TAP transport efficiency, and peptide binding to HLA. Moreover, there has been ample of data reported in IEDB (iedb.org) and as written in Table 2 and Table 3 on the manuscript, which show epitopes from ORF1ab gives positive results in the T-cell assays. This means that ORF1ab is a good immune antigen, able to induce T-cell responses. There are several other reasons that lead the authors to focus on ORF1ab as vaccine target, as written in paragraphs lines 981-1001. One of them is the fact that ORF1ab is the first protein to be synthesized by the infected cells, and therefore the epitopes from ORF1ab will appear earlier as compared to the epitopes from the proteins that synthesized later. The benefit of targeting the early epitopes is that it will allow T-cells to kill the infected cells before the virus has a chance to replicate.
Reviewer 2 Report
In this manuscript, Gustiananda and colleagues have determined a multi-epitope peptide-based vaccine that can cover a large human population and effective against any SARS-CoV-2 variants by identifying conserved and promiscuous CTL and HTL epitopes through immune-informatics prediction approaches. Overall, this manuscript is well written and prepared with ample informative in-silico data. The bioinformatics approach of current study sounds meaningful.
I have a couple of comments. What is the biological and clinical significance of data obtained from the current study to performing basic and translational research? What can be considered actually as the follow-up study towards SARS-CoV-2 vaccine design and development by using currently identified epitopes such as the highly promiscuous peptide 899WSMATYYLF907? The authors should address these matters more concretely.
Minor comments
- Overall English-editing needs to be done for correction of grammatical and typo errors in the text.
- Lines 3 and many other places: conserve -> conserved (?)
- Line 15: causes -> cause (?)
- Lines 25-26: able to induce -> be capable of inducing (?)
- Line 39: 4,2 million -> 4.2 million (?)
- Line 113: with not many -> with only small number of (?)
Author Response
Responses to Reviewer 2 comments
Major comments:
We thank Reviewer 2 for the valuable comments and suggestions with regard to the manuscript. We would like to address the comments on the biological and clinical significance of data by including this new paragraph into the conclusion section (Lines 1068-1096):
“……The current study generated data about SARS-CoV-2 ORF1ab T-cell epitopes and their characteristics such as the epitopes conservancy, HLA binding promiscuity, and the level of homology with peptides from human common cold coronaviruses, human self-proteins and microbiomes. Those characteristics are important for the development of a peptide-based vaccines which induce T-cell responses. T-cells target the antigens originated from all proteins of SARS-CoV-2, including ORF1ab. ORF1ab is intrinsically conserved because they are important for virus replication, therefore not easily mutated. Vaccines based on the evolutionarily stable protein is beneficial because it will work against all variants of SARS-CoV-2. The current study nominated 12 conserved and promiscuous epitopes to be used in the vaccine development that will cover majority of the Indonesian and the world population. One epitope in particular, 899WSMATYYLF907, was predicted to bind to HLA-A*24:07, which is the HLA allele predominant in Indonesian population.
We highlighted Indonesian in this study since the HLA background of the population is different from the Caucasian population as shown in Figure 3. HLA-A*24:07 is not very well studied and no data is available about SARS-CoV-2 T-cell epitopes as-sociated with this HLA (Table 3). The in silico data generated in this study should be followed by wet-lab experiments to map T-cell epitopes that will be recognized by COVID-19 convalescent individuals from Indonesia. Such study has not been done before, even though allele frequency for HLA-A*24:07 is significantly high (0.26) and that Indonesian (population of 277 million) is the 4th largest population in the world, that is also affected by the pandemic.
The study also generated other interesting findings related to the cross-reactive epitopes between SARS-CoV-2 and human proteins. Epitope 2784AIFYLITPV2792 matched with human peptide AIFYLITLV which is derived from the olfactory receptor. We hypothesized that epitope similarity might contribute to the anosmia symptoms in some COVID-19 patients. Experimental validation is needed to test the epitopes and the characteristics of T-cells recognizing the epitopes. The data generated might entangle the molecular mechanism of anosmia in some patients.….”
Minor comments
We thank Reviewer 2 for pinpointing some of the grammatical error in the text. We have addressed them and corrected them as follow:
Reviewer 2: Lines 3 and many other places: conserve -> conserved (?)
M Gustiananda: the words ‘conserve’ have been changed into ‘conserved’ in line 3 and many other places.
R2: Line 15: causes -> cause (?)
MG: the words ‘causes’ have been changed into ‘cause’
R2: Lines 25-26: able to induce -> be capable of inducing (?)
MG: line 25 change ‘able to induce’ into ‘capability of inducing’
R2: Line 39: 4,2 million -> 4.2 million (?)
MG: in line 38, the number ‘4,2 million’ has been changed into ‘4.2 million’
R2: Line 113: with not many -> with only small number of (?)
MG: in line 112 the words ‘with not many’ have been changed into ‘with only small number of’
Round 2
Reviewer 1 Report
Although I am still concerned that there is no experimental evidence to support this study, the author has already answered my questions. I think this study fits the scope of this journal and can be published.